

# How combining multi-scale monitoring and compound-specific isotope analysis helps to evaluate degradation of the herbicide *S*-metolachlor in agro-ecosystems?

Boris Droz[1,•*], Guillaume Drouin[1,•], Jenna Lohmann[1], Benoit Guyot[1], Gwenaël Imfeld[1], Sylvain Payraudeau[1*]

[1] Institut Terre et Environnement de Strasbourg (ITES), University of Strasbourg/ENGEES, CNRS UMR 7063, France
• These authors contributed equally to this work.

*Correspondence to*: Boris Droz (drozditb@oregonstate.edu) and Sylvain Payraudeau (sylvain.payraudeau@engees.unistra.fr)

**Abstract.** The presence of pesticides in surface water poses a significant risk to the quality of drinking water resources. A critical challenge in water quality management involves quantifying the export, degradation, and persistence of pesticides at the catchment scale. Compound-specific isotope analysis (CSIA) may help to evaluate the contribution of pesticide biodegradation in topsoil and water, as it is generally unaffected by non-degradative processes such as dilution, sorption, and volatilisation. In this study, multi-scale monitoring with CSIA was combined with a mass balance approach to determine the source apportionment and degradation contribution to the overall dissipation of *S*-metolachlor, a widely used herbicide, in the Souffel catchment (115 km²) during a corn and sugar beet growing season. The mass balance, including topsoil, river water, sediment, and wastewater treatment plant (WWTP) effluent, showed that $98.9 \pm 4.7\%$ ($\bar{x} \pm \mathrm{SD}$) of *S*-metolachlor applied during the study period was degraded over the five-month growing season. Most degradation occurred in the topsoil, with only $12.3 \pm 3.1\%$ degraded in the river. CSIA-based estimates of *S*-metolachlor degradation corroborated the mass balance results, indicating that $98 \pm 20\%$ of *S*-metolachlor was degraded over the growing season. WWTPs contributed to $52 \pm 18\%$ of the input mass based on daily discharges. However, *S*-metolachlor from non-point and point sources could not be clearly distinguished due to similar stable isotope signatures. Despite this limitation, our results demonstrate that pesticide CSIA, applied from upstream to downstream, enabled robust estimation of pesticide degradation across an entire catchment with relatively low sampling and analytical effort. We anticipate that CSIA will enhance surface water management by improving the diagnosis of pesticide off-site transport and degradation. This approach can support the development of efficient regulatory strategies aimed at preserving and restoring aquatic ecosystems.



**Short summary:**

How to evaluate pesticide persistence and degradation across an agricultural catchment to improve strategies for preserving aquatic ecosystems? By using a tracing method based on stable isotope signatures of pesticides at the catchment scale, degradation of the herbicide *S*-metolachlor could be distinguished from other dissipation processes. A limited number of isotopic measurements can provide critical insights for designing efficient strategies to protect aquatic ecosystems.

## 1 Introduction

At the European scale, the slightly upward trend in total pesticides sales from 2011 to 2018 highlights the increasing reliance of human activities on pesticides (European Environment Agency and Mourelatou, 2018), although a decrease of 12 % was observed from 2018 to 2022 (Eurostat, 2024). As a consequence of past and current pesticide use, widespread contamination affects all environmental compartments, including soils, surface water, groundwater and air (Tang et al., 2021). Between 2013 and 2022, 9% to 25% of monitored rivers and lakes in Europe reported pesticide concentration exceeding EU effect threshold (European Environment Agency, 2024). *S*-metolachlor, a herbicide, was among the three most frequently detected pesticides during this period (European Environment Agency, 2024).

Of all surface waters, low Strahler order rivers are among the most vulnerable aquatic systems to pesticide off-site transport from agricultural areas (Halbach et al., 2021; Spycher et al., 2018, Toth et al., 2024), as they are closely connected to their catchment (Engelhardt et al., 2014). Processes controlling pesticide loads entering rivers depend on transportation, as well as pesticide accumulation and degradation patterns, involving various compartments from the topsoil (i.e., 0 to 10 cm) to the river (Rasmussen et al., 2015). While biodegradation is recognised as the primary degradation process in agriculturally impacted catchments (Fenner et al., 2013), the interplay of hydrological dynamics and bio-physico-chemical processes on its contribution to overall pesticide dissipation, i.e., the observed concentration decline, remains poorly understood at the catchment scale. This uncertainty hinders the effective implementation of surface water protection measures, as apparent pesticide dissipation may obscure underlying persistence and accumulation at the catchment scale.

Hot spots of pesticide biodegradation at the catchment scale are primarily found in topsoils, surface waters, and the river hyporheic zone (Fenner et al., 2013). In a previous study, *S*-metolachlor biodegradation in topsoil was estimated to be as much as 93% three months after application (Alvarez-Zaldívar et al., 2018). Pesticide off-site transport to surface waters can occur in the dissolved phase via leaching or runoff, as well as in particulate form during intense rainfall event (Meite et al., 2018). The transport of particle-bound pesticides typically prolongs their transit time within the topsoil, thereby reducing their availability and the duration for biodegradation to occur (Menz et al., 2018). In addition, the hyporheic zone of rivers has been proposed as a highly reactive zone for contaminant biodegradation (Krause et al., 2017), leveraging redox and biogeochemical gradients generated by water flows and favoring microbial activities (Boano et al., 2014; Peralta-Maraver et al., 2018). Despite considerable efforts to evaluate dissipation processes at the catchment scale, tracking pesticide degradation under environmental conditions remains challenging due to limitations in current approaches. These approaches typically rely on





pesticide concentrations and the identification of transformation products to assess pesticide dissipation, covering both non-degradative dissipation processes and degradation across catchment compartments.

Compound-specific isotope analysis (CSIA) allows for the detection and potential estimation of contaminant degradation in the environment, although its application has not previously been employed to assess pesticide degradation in mid-scale agricultural catchments (from 50 to 500 km$^2$) (Elsner, 2010). Contaminant molecules containing lighter isotopes (e.g., $^{12}$C) typically undergo faster degradation relative to their heavier counterparts (e.g., $^{13}$C), which may result in a reaction-specific kinetic isotope effect. Consequently, the non-degraded fraction of the contaminant becomes enriched in the heavier isotopologues. Thus, monitoring changes of the isotope signature (e.g., $\delta^{13}C$) over time enable the evaluation of degradation and potentially facilitate the quantification of the degradation extent, provided that condition-specific reference isotopic fractionation values have been determined. CSIA has been utilised for the past two decades to assess the biodegradation of legacy industrial contaminants in groundwater (Hunkeler et al., 2008; Elsner and Imfeld, 2016). Recently, pesticide CSIA has been employed to examine degradation of the insecticide profenofos at the field scale (Masbou et al., 2025), the herbicide *S*-metolachlor in an small 47-ha agricultural catchment (Alvarez-Zaldívar et al., 2018) and the fungicide dimethomorph in a vineyard catchment (43 ha), from topsoil (Masbou et al., 2023) to the storm basin collecting runoff (Gilevska et al., 2023). Moreover, isotope mixing models based on CSIA data have been developed to aid in source apportionment at the hillslope scale (Lutz and Van Breukelen, 2014) and to predict pesticide biodegradation (Lutz et al., 2017). However, pesticide CSIA has not yet been utilised to aid in interpreting the dynamics of pesticide transport and dissipation processes within larger catchments (>100 km$^2$) that encompass river systems (Elsner and Imfeld, 2016).

This study investigated the transport and degradation of *S*-metolachlor from topsoil to the river network within a mid-scale (115 km$^2$) agricultural catchment. *S*-metolachlor, a widely used herbicide (Food and Agriculture Organisation of the United Nations, 2019), was selected as a model compound of synthetic pesticide due to its extensive application in corn and sugar beet crops until its recent ban in Europe (European Commission, 2024), as well as the persistence of its transformation products. The study aimed to (i) evaluate the potential of CSIA data collected along the river network as a proxy for evaluating upstream topsoil degradation of *S*-metolachlor, (ii) quantify the river network contribution to overall degradation at the catchment scale, and (iii) differentiate between pesticide sources, including diffuse agricultural applications and point-source inputs from wastewater treatment plants (WWTP). To achieve these objectives, a multi-scale sampling strategy was applied, integrating *S*-metolachlor mass balance, transit time analysis, and CSIA data from sources to the catchment outlet.

## 2 Material and methods

### 2.1 Catchment description

The Souffel catchment (115 km$^2$) is located northwest of Strasbourg (Bas-Rhin, France; catchment outlet coordinate: 48° 38' 20" N, 7° 44' 35" E; Fig. 1). The catchment has an average slope of 2.4 ± 2.2% ($\bar{x}$ ± SD; 0–86% min–max) with a total of 73 km of rivers comprising segments that range from the first to third order according to the Strahler classification system



(National dataset BD TOPO® 2019, http://www.ign.fr/). The mean annual precipitation recorded from 1975 to 2018 at the

nearest national weather station, located in Entzheim (13 km to the south; MétéoFrance, coordinates 48°32'24" N, 7°37'48" E), was 648 ± 91 millimetres. In the year of the study, 2019, the recorded precipitation was 579 millimetres. The distribution of crop types within this catchment is uniform, with cropland accounting for 84.5% of the total area (Corine Land Cover, 2012; v18.5.1; http://land.copernicus.eu). In 2019, the predominant crops were corn, wheat, and sugar beet, which occupied 41.4%, 20.6%, and 4.8% of the agricultural land, respectively (National Agricultural Plot Database – Registre Parcellaire Graphique,

www.ign.fr). Corn and sugar beet, under conventional farming practices, typically receive applications of herbicide products containing S-metolachlor (https://ephy.anses.fr, May 2019; see Table S1 in the Supplement).

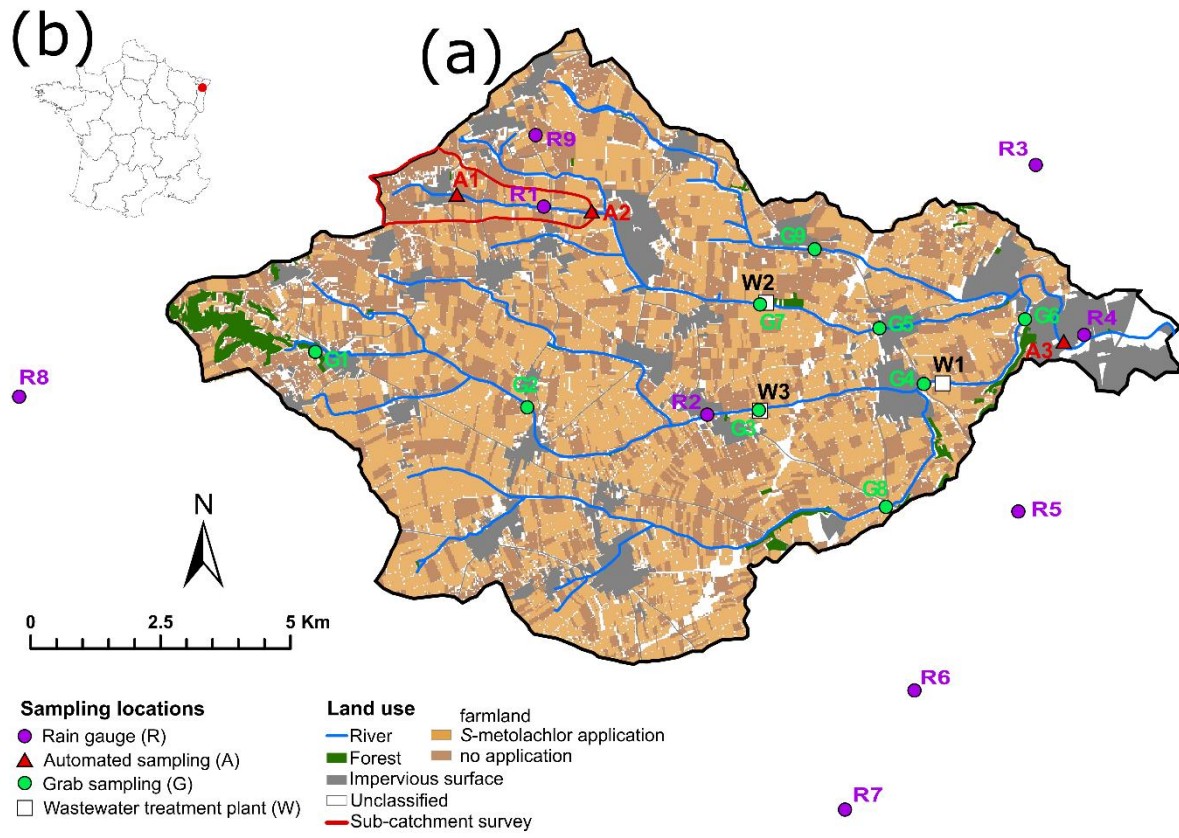

**Figure 1: The Souffel catchment (a), located in France (b), featuring sampling locations (monthly grab in green; n = 9 G1 to G9, high-resolution in red; n = 3 A1 to A3 and WWTP effluents in white; n = 3 W1 to W3) and the associated land uses. The area of S-**
**metolachlor applications in 2019 was estimated on crops with reported S-metolachlor authorisation (https://ephy.anses.fr/; corns, beets) using the Registre Parcellaire Graphique (www.ign.fr). Rainfall was quantified using nine rain gauges: R1 (deployed by our team for this study to characterise sub-hourly intensity), hourly stations R7 (MeteoFrance), R3-R6 (Strasbourg Eurometropole), and R2, R8 to R9 from citizen climate stations (www.wunderground.com). The targeted sub-catchment, Avenheimerbach river, is delineated in red.**



## 2.2 *S*-metolachlor applications

Determination of *S*-metolachlor applications rates at the catchment scale is crucial to establish the mass balance used to demonstrate the potential of combining multi-scale sampling and CSIA to estimate *S*-metolachlor degradation. Application estimates were derived from a survey conducted in the Avenheimerbach sub-catchment in 2019 (Fig. 1) and then extrapolated to the entire catchment, as detailed below. This sub-catchment is considered representative by local farming authorities due to its similar farming practices for corn and sugar beet compared to the entire Souffel catchment (Table S2 in the Supplement). The survey of pesticide application practices included the 60 farmers owning fields in the Avenheimerbach sub-catchment (3.6 km$^2$, containing 94% farmland; Fig. 1, red sub-catchment). The survey's responses covered 57% of the plots, with 43% potentially receiving *S*-metolachlor applications under reported homogeneous conditions. Two sequential applications of S-metolachlor were administered to sugar beet fields, with each application ranging from 576 to 672 grams per hectare, on April 18 (± 1 day) and April 29 (± 2 days). Subsequently, a single application of S-metolachlor, with doses varying from 160 to 1000 grams per hectare, was applied to corn fields between May 14 and June 3, with a median application date of May 20. These application rates and timings are consistent with the climatic conditions and vegetation stages observed in the Souffel catchment during 2019, as corroborated by local agricultural advisers and pesticide suppliers. Therefore, the *S*-metolachlor application amounts were extrapolated to the entire catchment for the calculation of the mass balance (Sect. S1.2 and S1.5 in the Supplement). Multiple scenarios for the application doses of *S*-metolachlor were evaluated, considering national regulations, recommendations from local farmer advisers, and survey results. This approach was employer to explicitly address uncertainty related to the extrapolation of application rates at the catchment scale (Table S1 in the Supplement). In total, an estimated 3163±116 kg of *S*-metolachlor (Table S10 in the Supplement) were applied across the study area during the study period.

## 2.3 Sampling schemes

The multi-scale data collection aims to: (i) characterise the main hydrological pathways from fields to rivers, (ii) quantify the extent of *S*-metolachlor degradation in topsoil and rivers, and (iii) determine *S*-metolachlor source apportionment, distinguishing between diffuse agricultural applications and WWTP point sources. The adopted strategy involved high frequency flow proportioned sampling at three nested scales (Fig. 1): the upstream (A1) and downstream (A2) parts of the first order Avenheimerbach sub-catchment, and the outlet of the catchment (A3). Additionally, a monthly sampling scheme was implemented to examine upstream-to-downstream river processes and differentiate contributions from multiple sources, including diffuse agricultural inputs and three-point sources corresponding to WWTP effluents (W1 to W3, Fig. 1). The sampling campaign was carried out over a period of 215 days, from March 1st to October 1st, 2019. This duration was chosen to encompass the application periods of S-metolachlor as well as the growing seasons for both sugar beets and corn. An additional dataset provided by the Rhin-Meuse Water Agency was used to support evidence of *S*-metolachlor degradation. This dataset included monthly measurements of three transformation products of *S*-metolachlor: metolachlor ethane sulfonic



acid (ESA), metolachlor oxanilic acid (OXA) and metolachlor NOA 413173, from January to December 2019. Water samples from eight monitoring sites (Fig. S8, in the Supplement) were analysed monthly by an ISO/IEC 17025-accredited laboratory (Eurofins Hydrologie Est, COFRAC) using validated methods, including NF EN ISO 11369 for pesticide residues (Table S13

in the Supplement). The data are expressed as *S*-metolachlor mass-equivalent loads ($MEL_{SM}$), calculated as shown in Equation 1.

$$MEL_{SM} = load(SM) + \sum_{i=1}^{TPs} load(TP_i) \times \frac{MolW_{SM}}{MolW_{TP_i}} \qquad (1)$$

where $MolW_{SM}$ and $MolW_{TPi}$ denote the molar weight of *S*-metolachlor, and that of its transformation products (TPs), respectively. The proportion of each TPs, i.e. *%ESA*, *%OXA* and *%NOA*, can be expressed as the ratio of the associated mass

equivalent loads on the sum of TPs mass equivalent loads and SM loads (Rose et al., 2018).



### 2.3.1 Continuous monitoring at three nested scales

Continuous flow proportional sampling was employed to quantify pesticide degradation at diverse sub-catchment size (outlets
at location A1, A2 and A3 in Fig. 1, respectively 2.3, 3.6 and 115 km²) between March 1 and October 1, covering 215 days.
This approach enabled the estimation of representative weekly average concentrations and loads, which were used to calculate
seasonal mass balances and export coefficients. The autosamplers, flowmeter and multiparameter probe deployed at each
location are detailed in the Supplement (Table S3). Discharge, with an average precision of 15%, along with $pH$, conductivity
and water temperature, were recorded continuously at a 2 minute frequency at the outlet A1, A2 and A3. Samples were
160 collected over defined discharge volumes, adjusted weekly, from 840 to 1860 m³ at locations A1 and A2, and from 1500 to
1960 m³ at A3, to account for the seasonal decline in baseflow. Water samples (up to twelve 330 mL glass bottles) were
collected weekly and merged into composite samples based on hydrograph characteristics, separating baseflow and high-flow
conditions. Hourly rainfall data were collected in nine rain gauges (Fig. 1; ITES, MétéoFrance, Strasbourg Eurometropole and
open-source data, www.wunderground.com) covering the Souffel catchment area (R1 to R9, Fig. 1 and Table S4 in the
165 Supplement). Rainfall events were separated by at least 24 hours of no rainfall and classified using R-v.3.5.0 with the hydromad
package v.0.9-27 (Andrews et al., 2011).

### 2.3.2 Monthly sampling of water, sediment and topsoil

Eleven sampling locations for water and riverbed sediment were selected along the Souffel river (G1 to G9 and A1 to 3 in Fig.
1). Site G1 was near the primary source, characterised by a land cover of 32% forest and 68% cropland. Sites G2 to G9 were
170 selected using a stratified random sampling approach within the agricultural areas to ensure a homogeneous cropland cover
ranging from 84 to 94%. Additionally, WWTPs within the catchment area (W1 to W3 in Fig. 1) were sampled as potential
secondary sources of $S$-metolachlor, likely originating from various pathways, including infiltration from crop topsoil into
underground sewage networks, agricultural runoff, household usage and improper disposal during pesticide preparation or
equipment cleaning (Gerecke et al., 2002). Grab samples of river water and wastewater treatment plant (WWTP) effluent were
175 collected on a monthly basis, specifically on March 13, April 2, May 15, June 12, July 17, August 21, and October 1. These
samples were gathered using 2-liter high-density polyethylene (HDPE) bottles and subsequently analysed for S-metolachlor
concentration and compound-specific isotope analysis (CSIA). In addition, 150 mL samples were collected in HDPE bottles
and subjected to on-site filtration using 0.45 µm cellulose acetate filters for further hydrochemical analysis (Table S5 in the
Supplement). At each location, conductivity, $pH$, water temperature and flow were measured in the river (Table S3 in the
180 Supplement).

Sediment samples were collected monthly from the riverbed at sites G1, G3 to G9 and A1 to A3 (Fig.1) to evaluate whether
the sediment acted as a sink or source of $S$-metolachlor. The top five centimeters of the riverbed sediment were collected using
a clean scraper and transferred into plastic bags. Sampling at G2 was not possible due to a gravel riverbed. Additionally, topsoil
samples (0 to 10 cm) were collected from sugar beet and corn fields at two locations within the surveyed sub-catchment (within



185 a 100 m radius of sites A1 and A2 in Fig. 1) to quantify *S*-metolachlor dissipation over the season. At each location, four subsamples were combined and thoroughly mixed. All water, sediment and topsoil samples were stored in dark, sealed containers at 4 °C and processed for *S*-metolachlor extraction within two days of collection.

**2.4 *S*-metolachlor extraction, quantification, and compound-specific isotope analysis (CSIA)**

Water samples from continuous and grab sampling were initially pre-filtered using qualitative filters (Grade 1, Whatman) and
190 subsequently filtered through glass filters (GF/5, 0.4 μm average pore size, Macherey-Nagel) with a bottle-top vacuum filtration unit. The resulting filtrate was extracted and concentrated by solid-phase extraction (SPE) using SolEx C18 cartridges (1 g phase, 40-63 μm particle size, Thermos Fischer Scientific). The filters were initially collected and subsequently dried at ambient temperature within a desiccator to ascertain the total suspended solids (TSS). Following this, the filters were frozen prior to the extraction of S-metolachlor.

195 To quantify solid-bound *S*-metolachlor, 34% of the filters (n = 93) were selected based on the highest concentrations observed in the aqueous phase (≥0.02 μg L$^{-1}$) and the mass of TSS retained on the filter (≥13 mg L$^{-1}$). *S*-metolachlor was extracted from these filters by solid-liquid extraction, as previously described (Gilevska et al., 2022). Topsoil and wet riverbed sediment samples were homogenised and sieved through a 2 mm mesh, followed by centrifugation at 2400 relative centrifugal force (RCF) for 20 minutes to eliminate excess water. Subsamples of the topsoil and wet riverbed sediment were collected for further
200 hydrochemical analysis (refer to Table S5 in the Supplement). Approximately 20 grams of dried mass equivalent from the wet sediment or topsoil samples underwent solid-liquid extraction, following the same procedure used for TSS.

*S*-metolachlor concentrations were quantified using a validated gas chromatography–mass spectrometry (GCMS) method (Droz et al., 2021). The environmental quantification limits ranging from 0.01 to 0.03 μg L$^{-1}$ for water and 0.4 to 2.5 μg kg$^{-1}$ for riverbed sediment or topsoil, with an analytical reproducibility of 4.3%.

205 Compound-specific isotope analysis (CSIA) of carbon in S-metolachlor was conducted in accordance with established protocols for gas chromatography–isotope ratio mass spectrometry (GC-IRMS) (Gilevska et al., 2022). In brief, the isotope ratios $^{13}C/^{12}C$ for each sample were measured in triplicate, with a typical analytical uncertainty of ±0.5‰. These measurements are expressed as delta values $\delta^{13}C$ relative to the Vienna PeeDee Belemnite (V-PDB) standard:

$$\delta^{13}C = \frac{R(^{13}C/^{12}C)_{sample}}{R(^{13}C/^{12}C)_{standard}} - 1 \qquad (2)$$

210 where $\delta^{13}C$ represents the isotope signature of the carbon and $R(^{13}C/^{12}C)$ is the isotope ratio of $^{13}C$ to $^{12}C$ isotopes in a given sample or a standard. The $\delta^{13}C$ linearity is defined as the range of injected carbon mass within which the $\delta^{13}C$ values of *S*-metolachlor remain consistent within ± 0.5 ‰ (Jochmann et al., 2006), which was established from 6 to 300 ng for C. The minimal detectable change of isotope signature ($\Delta\delta^{13}C_{min}$), above which isotope fractionation can be attributed to degradation, is determined by propagating the uncertainties associated with measurements and sample preparation (Eq. S1 in the
215 Supplement). For *S*-metolachlor, $\Delta\delta^{13}C_{min}$ was 1‰ in water and 2‰ in sediment. However, as the origin of the isotope



fractionation cannot be identified a priori in environmental samples, a conservative threshold of 2‰ was applied for all samples.

**2.5 Data analysis**

**2.5.1 *S*-metolachlor loads and mass balance**

The *S*-metolachlor loads were estimated at the sub-catchment and catchment outlets (A1 to A3 in Fig. 1) on a daily time-step using Eq. 3:

$$M_{A_{j,i}} = \int_{i-1}^{i} Q_{A_j}(t) \times C_{A_{j,i}} \tag{3}$$

where $M_{A_{j,i}}$, $Q_{A_j}(t)$ and $C_{A_{j,i}}$ represent the exported load, the instantaneous water discharge, and the average *S*-metolachlor concentration at station $A_j$ during period $i$, respectively. The *S*-metolachlor concentrations in composite water samples were assumed to be representative average values for the water collection period. Area-normalised water daily discharges (m³ s⁻¹ ha⁻¹) and *S*-metolachlor loads (g ha⁻¹) (Shaw et al., 2019) were used to compare *S*-metolachlor exports at sites A1 to A3. Due to missing water samples or flow measurements caused by acquisition failure following extreme floods or severe weather events throughout the agricultural season, continuous flow and concentration time-series covered 62%, 59% and 76% of the total monitoring period at sites A1, A2 and A3, respectively. Missing data at sites A1 and A2 were reconstructed using the upstream-downstream relationship described in the Supplement (Fig. S3).

Daily loads of S-metolachlor were computed for each grab sampling event conducted in the river at sites G1 to G9, as well as for the effluents from WWTP (W1 to W3). This calculation was performed by multiplying the instantaneous concentrations of S-metolachlor by the corresponding discharge rates. Discharge in the river was measured using a handheld electromagnetic water flow meter, while discharge data for WWTP effluents were provided by the respective WWTP managers. For the three river reaches receiving WWTP effluents (upstream locations G3, G4 and G7), source apportionment of *S*-metolachlor was estimated using mass mixing (Eq. 4). The approach combined the daily contributions from the WWTPs ($L_{wwtp}$) and the upstream river ($L_{river}$) to determine the contribution of the WWTP ($x_{wwtp}$) to the total *S*-metolachlor load in the river reach.

$$x_{wwtp} = \frac{L_{wwtp}}{L_{wwtp} + L_{river}} \tag{4}$$

At the outlet A3, mass conservation was assumed to estimate the contribution of point and non-point sources to the total *S*-metolachlor load. A seasonal mass balance, detailed in the Supplement (Sect. S1.5), was calculated over the 215-day study period and compared with CSIA data. The analysis accounted for relevant processes occurring in the catchment, as described by Eq. 5:

$$m_{app} + m_{stock\ y-1} + m_{wwtp} = m_{exp} + m_{vol} + m_{photo} + m_{hydro} + m_{wat,bio} + m_{soil,bio} + m_{res} \tag{5}$$



where $m_{app}$ represents the pesticide mass applied across the catchment in 2019, based on the application scenarios detailed in Table S1 in the Supplement. $m_{stock\ y-1}$ denotes the residual pesticide stock present in the 2018 topsoil prior to the first *S*-metolachlor application in 2019. $m_{wwtp}$ accounts for the *S*-metolachlor effluent discharged from WWTPs, while $m_{exp}$ represents the exported load measured at the catchment outlet (A3, Fig. 1). The terms $m_{vol}$ and $m_{photo}$ correspond to the estimated masses lost due to volatilization and photodegradation, respectively, with detailed assumptions and references provided in Sections S1.5.1 and S1.5.2 of the Supplement. $m_{hydro}$ represents the mass degraded by hydrolysis, which was considered negligeable for *S*-metolachlor under the study field and river conditions (Masbou et al., 2018). Additionally, $m_{wat,bio}$ and $m_{soil,bio}$ represent the *S*-metolachlor masses biodegraded in river and agricultural fields, respectively. The pseudo first-order rate constants for *S*-metolachlor have been previously determined for water-sediment systems (Droz et al., 2021) and agricultural topsoil (Alvarez-Zaldívar et al., 2018). Finally, $m_{res}$ refers to the residual *S*-metolachlor measured in topsoil after the 215-day study period.

**2.5.2 Degradation estimation and source apportionment using CSIA**

The extent of *S*-metolachlor degradation ($B$) in river and WWTP water samples was determined using carbon isotope signatures ($\delta^{13}C$) through the CSIA approach (Eq. 6) (Alvarez-Zaldívar et al., 2018). The carbon isotopic fractionation for *S*-metolachlor ($\varepsilon_{bulk,C} = -1.2 \pm 0.4$‰) was derived from laboratory-controlled biodegradation experiments conducted under aerobic conditions (Droz et al., 2021).

$$B = 1 - \left(\frac{\delta^{13}C(t)+1}{\delta^{13}C_0+1}\right)^{1/\varepsilon_{bulk,C}} \tag{6}$$

Estimating *S*-metolachlor degradation in the topsoil of corn and sugar beet fields was hindered by strong matrix effects, which compromised reliable $\delta^{13}C$ measurements. Consequently, $\delta^{13}C$ dynamics and degradation extent were inferred from concentration data (Section 2.3.2) using a degradation model (Payraudeau et al., 2024). The model, linking dissipation, degradation, and stable isotope fractionation of *S*-metolachlor, was developed and successfully validated in a nearby agricultural catchment with comparable soils and farming practices (Sect. S1.6 in the Supplement).

The utility of CSIA data for distinguishing point and non-point sources of *S*-metolachlor was evaluated using an isotope mixing approach (Eq. 7):

$$\delta^{13}C_{outlet} = x_{wwtp} \times \delta^{13}C_{wwtp} + (1 - x_{wwtp}) \times \delta^{13}C_{pred,river} \tag{7}$$

where $\delta^{13}C_{outlet}, \delta^{13}C_{wwtp}, \delta^{13}C_{pred,river}$ represent the isotope signatures measured at the outlet (A3), in the WWTP effluents, and the predicted signatures in the river from upstream agricultural areas, respectively. The term $x_{wwtp}$ denotes the proportion of the daily *S*-metolachlor load at the outlet originating from WWTP effluents. The proportion $x_{wwtp}$ was calculated using a stable isotope mixing model (Eq. 8).



$$x_{wwtp} = \frac{\delta^{13}C_{outlet} - \delta^{13}C_{pred,river}}{\delta^{13}C_{wwtp} - \delta^{13}C_{pred,river}} \qquad (8)$$

This mixing model was applied exclusively to CSIA data exhibiting significant degradation ($\Delta\delta^{13}C \geq 2‰$).

## 3 Results and discussion

The predominant hydrological processes governing the off-site transport of *S*-metolachlor from topsoil to river systems are first summarised. Subsequently, dissipation and export patterns of *S*-metolachlor were analysed by integrating multi-scale sampling with a mass balance approach. Finally, *S*-metolachlor degradation along the topsoil-to-river continuum was estimated

using CSIA data, with a focus on: (1) the use of stable isotope ratios of *S*-metolachlor as an indicator of degradation in upstream topsoil, (2) the specific contribution of the river to the overall catchment-scale degradation, and (3) the apportionment of multiple pesticide sources, including diffuse agricultural applications and point sources from WWTPs.

### 3.1 Dominance of lateral subsurface flows in *S*-metolachlor load and significance of WWTPs contribution

Hydro-climatic patterns and key hydrological processes potentially driving pesticide off-site transport were characterised using

a sampling approach that combined upstream to downstream monitoring and sampling at monthly to sub-hourly time scales. Compared to the previous twenty years (March to October 2000 to 2019), 2019 for the same month was the five time drier and three times warmer, with an area-normalised discharge at the outlet (A3) of $0.828 \pm 0.986$ m$^3$ day$^{-1}$ ha$^{-1}$ ($\bar{x} \pm$ SD; Fig. 2e) and an average temperature of $15.7 \pm 0.7$ °C ($\bar{x} \pm$ SD), respectively. This is one order of magnitude lower than that of the neighboring rivers to the south and the north of the study area during the study period (6.55 m$^3$ day$^{-1}$ ha$^{-1}$ for the Bruche River

and 3.85 m$^3$ day$^{-1}$ ha$^{-1}$ for the Zorn River; http://www.hydro.eaufrance.fr). This difference is consistent with longer-term data (1980 to 2000; http://www.hydro.eaufrance.fr), and highlights the limited capacity of the Souffel catchment in 2019 to dilute pesticide loads compared to neighbouring rivers with mountainous upstream areas. Area-normalised discharges calculated at grab sampling locations (G1 to G9 and A1 to A3; Fig. 1) were close to the 1:1 line (Fig. S3 and S4 in the Supplement). This suggests homogeneous hydrological responses and water distribution (Shaw et al., 2019) between the eleven sub-catchments

and associated river reaches. No significant differences of the river hydrochemistry were observed between locations and sampling times (Table S11 in the Supplement; Tukey's test; *p*–value >0.05), although WWTP hydrochemistry significantly differed from river samples (Sect. S2.6 in the Supplement). The three WWTP effluents contributed to $49 \pm 6\%$ of the total water discharge at the catchment outlet during the high-flow period (March to June), and up to 100% during the low-flow period (July to September; Fig. S6a in the Supplement).

The hydrological response time of the catchment area, defined as the time between the peak of a rainfall event (Fig. 2a) and the peak of the associated discharge (Fig. 2e) at the main outlet A3, ranged between 1.2 and 12 hours. This suggests the co-occurrence of both rapid runoff and slower sub-surface contributions to the river (Gericke and Smithers, 2017). The hydro-climatic dataset was also used to investigate the contributions of these fast flow processes. Seventeen rainfalls events were





recorded over the 215 days of the study (Fig. 2a, March to October 2019), with a cumulative rainfall of $358 \pm 50$ mm ($\bar{x} \pm$ SD
between stations R1 to R9).

The intensity of rainfall and the saturated hydraulic conductivity of the topsoil were compared for 17 rainfall events to estimate the likelihood of surface runoff. Between March 1 and October 1, only events 9 and 15 were classified as heavy rainfall (>7.5 mm h$^{-1}$) (Monjo, 2016), while 14 events were categorized as light (<2.5 mm h$^{-1}$), and one event as moderate ( 2.5–7.6 mm h$^{-1}$). Although events 8 and 12 exhibited high intensity over short period (Fig. 2a), their total rainfall depths were low.
Compared to the estimated saturated hydraulic conductivity of the topsoil, ranging from 6 to 60 mm h$^{-1}$ during corn and sugar beet growing seasons in a similar brown and calcic soil catchment (Lefrancq et al., 2017b), only events 9 and 15 were likely to generate moderate surface runoff. This suggests that preferential flow paths in the subsurface were likely the dominant transport mechanism for the other rainfall events. Furthermore, the absence of correlation between TSS concentration at the outlets (A1, A2, and A3) and rainfall events supports this hypothesis. Average TSS concentrations were $3,663 \pm 9,745$, $1,401$
$\pm 2,578$ and $578 \pm 938$ mg L$^{-1}$ ($\bar{x} \pm$ SD) at A1, A2, and A3, respectively, with maximum values of 49,605, 10,998 and 5,967 mg L$^{-1}$ (Fig. 2b and 2e). The lower TSS concentrations associated with subsurface flow compared to surface runoff further reinforce this hypothesis. Additionally, the limited extent of contributing surface runoff areas, as illustrated by the Wetness index (Sect. S2.2 and Fig. S5 in the Supplement), supports the hypothesis that subsurface flow may be the primary contributor to discharge at the outlet.
At the outlet (A3), lower electrical conductivity values were consistently observed during rainfall events, with an average drop from $0.852 \pm 0.097$ to $0.133 \pm 0.045$ mS cm$^{-1}$ ($\bar{x} \pm$ SD). Consequently, discharge at the outlet was inversely correlated with conductivity ($R^2 = 0.53$; p-value <0.01). This suggests that river water during the wet season primarily originated from subsurface flow, as rainwater ($\sigma$ <0.03 $\pm$ 0.01 mS cm$^{-1}$) entering the river likely had limited contact time with topsoil (Leu et al., 2004). Combined with earlier analysis showing that rainfall intensity rarely exceeded topsoil hydraulic conductivity,
indicating limited surface runoff, these results suggest that rapid transport from fields near the A1–A2 river reach (2.2 km) was primarily driven by lateral subsurface flow. Consistent with observations by Rose et al. (2018) in subsurface flow-dominated catchments, the seasonal ESA to OXA ratio of 3.1, corresponding to mean values of $56.8 \pm 12.3\%$ for ESA and $16.7 \pm 4.4\%$ for OXA across the eight Water Agency sites (n = 88, Tab. S12 in the Supplement), further supports the predominance of subsurface flow in the hydrological functioning of the Souffel catchment. During dry periods, the higher
conductivity was observed at A2 ($\sigma$= 1.19 $\pm$ 0.25 mS cm$^{-1}$) compare to A1 ($\sigma$ = 0.58 $\pm$ 0.31 mS cm$^{-1}$; $\bar{x} \pm$ SD), suggests groundwater mixing with river water. Groundwater conductivity measured at piezometers located 2 m from the left and right riverbanks in A2 on October 1, 2019, was $\sigma$ = 1.33 $\pm$ 0.06 mS cm$^{-1}$ (Sect. S1.7 and Fig. S2 in the Supplement), further supporting the effect of groundwater contributions during dry periods.





**Figure 2: Temporal dynamics of rainfall, river discharge, suspended solid and *S*-metolachlor concentration at the outlet of the sub-catchment A1, A2 (left panel) and Souffel catchment A3 (right panel) in 2019. (a) Rainfall at location R1 (Fig. 1), (b & e) River discharge at A1 & A2 and total suspended solids at A3, (c & f) *S*-metolachlor concentrations in dissolved phase ($n_{A1}$ = 58; $n_{A2}$ = 68; $n_{A3}$ = 176) and (d & g) Cumulative area-normalised load in dissolved phase. Dashed vertical line in (b & e) represents the timing of the seven grab sampling campaigns. The shaded gray area in (c, d, f, g) indicates the dates of *S*-metolachlor application.**



**3.2 _S_-metolachlor dissipation and mass export patterns**

In addition to transit time, both horizontal and vertical hyporheic exchange flows can affect the dissipation and biodegradation of _S_-metolachlor at the river scale by enhancing the mixing of nutrients and _S_-metolachlor in the water column with microbial

degraders in the riverbed sediments (Schaper et al., 2018). Although quantifying hyporheic exchange along the 73 km of river reaches was beyond the scope of this study, multiple lines of evidence indicate that such flows were limited in the study system. First, the riverbed sediment was classified a silt loam (Gerakis and Baer, 1999), with a composition of clay (<2 μm) at $14.6 \pm 3.5\%$ ($\bar{x} \pm$ SD), silt (2 to 50 μm) at $65.5 \pm 11.5\%$, and sand (50 to 2000 μm) at $19.9 \pm 13.7\%$ (n = 22), and had a low organic carbon content ($C_{org} = 2.6 \pm 1.3\%$). This composition indicates low permeability ($K_s < 10^{-6}$ cm s$^{-1}$) (Ren et al., 2018), which

limits hyporheic exchange flows (Kunkel and Radke, 2011). Second, _S_-metolachlor was detected in only 10% of riverbed sediment samples, with low concentrations (<12 μg kg$^{-1}$) throughout the sampling season, indicating that sediments were not a significant source or sink for the water column. Regular grab samples revealed that sediments became anoxic within a few centimetres, potentially reducing biodegradation rates by up to a factor of seven (Droz et al., 2021). This is consistent with observations during low-discharge periods (July to October), when the water mass balance between the two river branches

(A1 and A2) and their confluence was unequal (Fig. S3, Supplement), and the combined daily discharge from upstream WWTPs exceeded the outlet discharge (Fig. S7, Supplement). During these low-flow periods, an estimated $10 \pm 10\%$ of the river volume infiltrated through the riverbed, a process known to limit sediment reactivity (Kunkel and Radke, 2011). Furthermore, the Souffel River's low turbulence during baseflow, lack of channel tortuosity, waterfalls, or dams, and generally low discharge further constrain hyporheic exchange flows (Voermans et al., 2018). Consequently, the potential for _S_-

metolachlor biodegradation associated with hyporheic exchange flows is considered negligible.

Throughout the five-month growing season, _S_-metolachlor was quantified in 88% of the composite samples collected along the river from upstream to downstream (dissolved phase, $n = 254$; Fig. 2c; detailed per location in Table S8 in the Supplement), including at the outlets A1 to A3 and grab sampling locations G1 to G9. Dissolved-phase concentrations ranged from 0.02 to 54.6 μg L$^{-1}$, with an average of $1.13 \pm 4.28$ μg L$^{-1}$ ($\bar{x} \pm$ SD; Fig. 2f, at A3). In the particulate phase, _S_-metolachlor

concentrations were below the detection limit for all samples (1.2 μg kg$^{-1}$ TSS; $n = 86$). _S_-metolachlor was quantified in 11% of sediment samples ($n = 76$), with concentrations ranging from 1.7 to 11.9 μg kg$^{-1}$. Multi-scale sampling revealed low background concentrations in the river before _S_-metolachlor application in April and May 2019 (from March to early April: $0.1 \pm 0.3$ μg L$^{-1}$; $\bar{x} \pm$ SD; n = 30), suggesting substantial dissipation of _S_-metolachlor applied during the previous 2018 season. Monthly comparisons of upstream and downstream _S_-metolachlor concentrations from April to October 2019, across the 11

river reaches (A1 to A3 and G1 to G9), revealed three distinct patterns (Fig. 3a). In 90% of cases ($n = 59$) concentrations did not vary significantly between upstream and downstream, as underlined by the 1:1 slope (Fig. 3a). This suggests a homogeneous distribution of _S_-metolachlor across the Souffel catchment (Gericke and Smithers, 2014), with no significant in-river dissipation.





A significant increase in *S*-metolachlor concentrations (95% confidence interval) was observed at specific river reaches and sampling dates (n = 5). Two key events were identified: (i) diffuse agricultural applications in April led to a marked concentration increase between the upstream site (G1) and its downstream counterpart (G2), and (ii) point-source inputs from WWTP discharges were detected on April 2 (Fig. 3a). In contrast, two instances ($n = 2$) showed significant decreases in *S*-metolachlor concentrations, specifically between G2 and G3 on April 2 and between G9 and the outlet in June.

Overall, seasonal dynamics of *S*-metolachlor from point and non-point sources highlighted its transport within the catchment. Mass contributions from WWTP effluents ($x_{wwtp}$; Eq. 4) and upstream non-point agricultural sources ($1-x_{wwtp}$) were estimated using daily WWTP discharge data and *S*-metolachlor concentrations. WWTP discharges accounted for the majority of pesticide load over short periods, while non-point source contributions from topsoils toward rivers were smaller but more consistent. The proportion of WWTP-derived *S*-metolachlor ($x_{wwtp}$) varied throughout the season (Fig. S6b in the Supplement), ranging from 0 to 100% of the observed mass load at the catchment outlet, with an average contribution of 53% from March to June. This aligns with a previous study at WWTP W1, which reported an $x_{wwtp}$ at 53 ± 24%, with up to 100% mass contribution on certain days during the 2015 to 2016 period (Agence de l'Eau Rhin-Meuse, 2020). Consequently, it was estimated that approximately 9% of the total off-site *S*-metolachlor export during rainfall events 1 and 2 in April 2019 (Table S9) originated from the WWTPs. Several sources may explain the presence of *S*-metolachlor in the WWTPs. *S*-metolachlor could originate from: (i) water infiltration from crop topsoil into the underground sewage network or ditch system (Wendell et al., 2024), (ii) agricultural runoff, which carries pesticides from crop topsoil to stormwater that eventually reaches the WWTP (Sutton et al., 2019), (iii) household usage, which is not approved in Europe (European Commission, 2016), or (iv) releases from the sewage system during pesticide preparation or sprayer clean-out at farmyards lacking properly regulated closed-loop washing stations (Le Foll et al., 2017). The timing of *S*-metolachlor detection in the WWTP effluent coincided with its application at the catchment scale. Although a diagnosis of the wastewater network by the WWTP manager (personal communication) supports the fourth hypothesis as the most probable source, a other potential sources cannot be excluded without further investigation.

From March 1 to October 1, low mass export coefficients of dissolved *S*-metolachlor were observed at catchment outlets A1, A2, and A3. These coefficients corresponded to 0.09% (25.6 ± 4 g) at A1, 0.06% (39.6 ± 3 g) at A2 (Fig. 2d) and 0.04 to 0.12% at A3 (4.87 ± 1.00 kg, Fig. 2g) (Table S1 and S9 in the Supplement). These results aligned with the 2019 hydro-climatic patterns (Sect. 3.1), which indicated light to moderate rainfall intensities and lateral subsurface flows as the predominant transport mechanism, with occasional surface runoff events. The observed export coefficients fall within the range of previous reported *S*-metolachlor concentration during entire agricultural seasons (0.072% to 8%) (Boithias et al., 2011; Alvarez-Zaldívar et al., 2018; Rose et al., 2018; Lefrancq et al., 2017a). Such low coefficients are consistent with the slow transfer of *S*-metolachlor from topsoil to the river via subsurface flow, as suggested in section 3.1 and previously demonstrated in a nearby catchment.




Although subsurface flow was the primary driver of *S*-metolachlor export, notable export events of *S*-metolachlor were related to post-application rainfall events. Combined rainfall events 3 and 4, as well as rainfall event 5, accounted for 21% and 47%, respectively, of the total export at outlet A3 (Fig. 2g). Notably, the highest export occurred during the low-intensity rainfall event 5 ($1.6 \pm 0.2 \ \mathrm{mm \ h^{-1}}$). During events 1 to 7 (up to June 15), *S*-metolachlor load dynamics were transport-limited,

accounting for 91% of the seasonal load. Thereafter, a shift to mass-limited source dynamics was observed, consistent with previous studies (Peter et al., 2020; Fairbairn et al., 2016). Overall, *S*-metolachlor transport at the catchment scale was mainly governed by the application dose and timing, its dissipation in the topsoil, and prevailing hydro-climatic conditions.

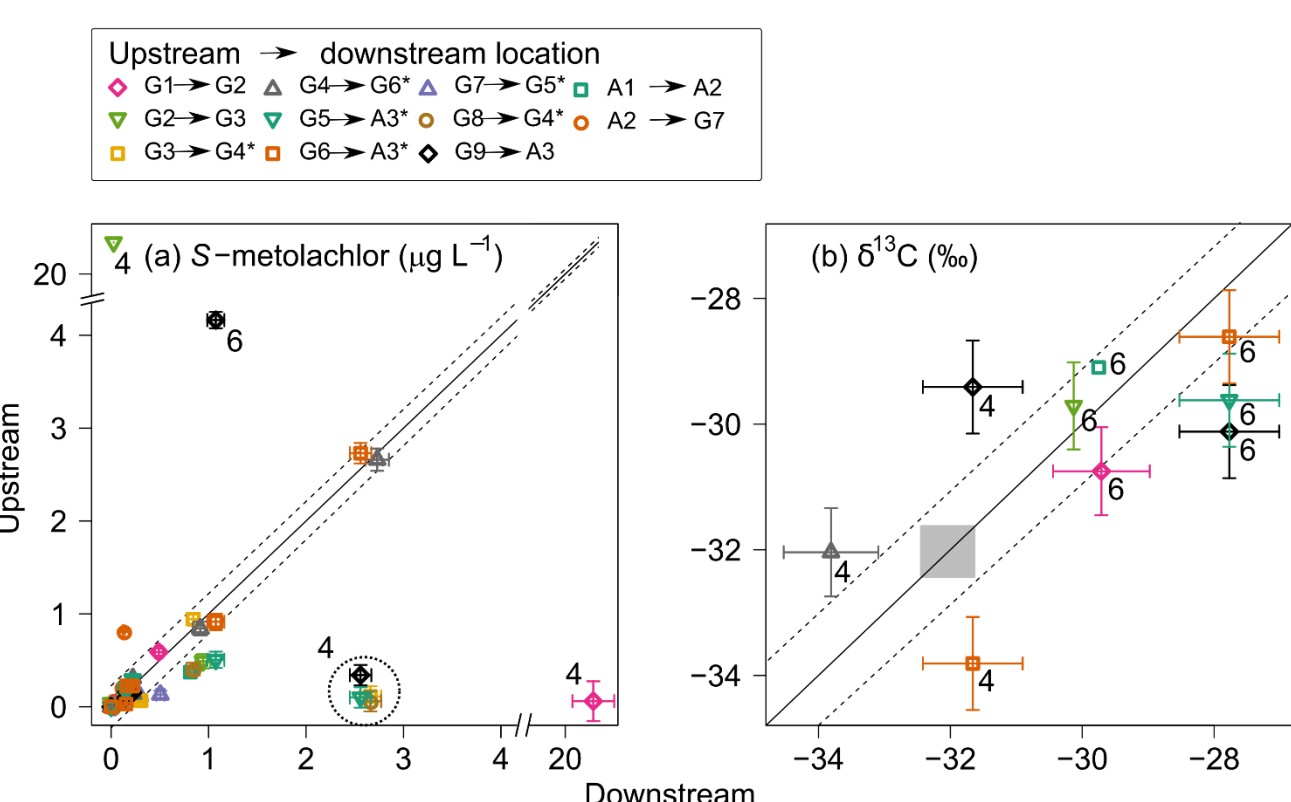

**Figure 3: Dynamic of *S*-metolachlor from upstream to downstream for the river reaches. *S*-metolachlor (a) concentrations and (b) carbon stable isotope composition ($\delta^{13}$C) in April (label 4) and June (label 6). The black solid line illustrates a 1:1 relationship, whereas the dashed line signifies the 95% confidence interval associated with the measurements. The grey zone illustrates the initial isotope composition of *S*-metolachlor commercial formulation ($\delta^{13}C_0$ = –31.8 ± 0.3‰). Error bars denote the propagated uncertainty from triplicate measurements for both concentration and $\delta^{13}C$. Asterisks (\*) denotes reaches receiving WWTP effluent.**



As demonstrated above, the combination of discharge data, hydrochemistry and *S*-metolachlor patterns from multi-scale sampling revealed the primary processes governing off-site transport and enabled the estimation of *S*-metolachlor dissipation and export coefficients. To construct the mass balance of *S*-metolachlor at the catchment scale from the application date to October 2019 (Sect. S2.5 in the Supplement), data from previous laboratory work (Drouin et al., 2021; Droz et al., 2021) were
425 integrated with soil and river sampling results (Eq. 5; Table S10 in the Supplement). The resulting mass balance, which incorporated inputs from soil, river water, sediment, and WWTP effluent, was nearly closed (<2% discrepancy). It indicated that $98.9 \pm 4.7\%$ (mean $\pm$ SD) of the applied *S*-metolachlor was degraded over the season, based on uncertainty scenarios related to application estimates (Sect. 1.2). Although constructing such a mass balance and quantifying pesticide degradation at the catchment scale is feasible, it demands extensive analytical and sampling efforts, which considerably limit its routine
application by water and territorial management agencies. The following section presents additional insights into *S*-metolachlor degradation derived from combining CSIA with the multi-scale sampling approach at the catchment scale.

### 3.3 *S*-metolachlor degradation from topsoil to river outlet and source apportionment using CSIA

The isotope signature of *S*-metolachlor observed in the river reflects degradation processes occurring in both agricultural topsoil and the river itself. While the extent of *S*-metolachlor degradation in topsoil can be estimated from direct measurements
of its isotopic signature, using the concentration change model described in Section S.1.6 of the Supplement, assessing degradation in the river water phase requires multi-scale monitoring. The topsoil model integrates observed dissipation, degradation, and shifts in the carbon stable isotope ratios of *S*-metolachlor. Predicted concentrations closely matched those measured in topsoil samples collected between March and October (Fig. S1, Supplement), indicating rapid degradation in 2019, with over 80% of the applied *S*-metolachlor degraded within the first two months post-application. These findings are
consistent with observations from a nearby agricultural catchment (Alvarez-Zaldívar et al., 2018). The predicted $\delta^{13}C$ isotope signatures from topsoil followed a clear seasonal trend post-application (Fig. 4), contributing to the assessment of topsoil degradation at the catchment scale.

Assuming constant lateral inputs of *S*-metolachlor on a daily timescale, comparing predicted $\delta^{13}C$ in topsoil with observed $\delta^{13}C$ values in the river from upstream to downstream allowed for quantification of in-river *S*-metolachlor degradation.
However, due to a minimum carbon mass required for accurate GC-IRMS analysis ($\geq$20 ng of carbon for *S*-metolachlor), only a subset of dissolved water samples (Fig. 4) was measurable. Results indicated that in-river *S*-metolachlor degradation between upstream and downstream locations was below the sensitivity of the CSIA method ($\delta^{13}C = 1‰$, <50% degradation). Consequently, the isotope signature at the outlet primarily reflects topsoil degradation at the catchment scale. This assumption was supported by a mass balance approach (Table S10 in the Supplement), which estimated $13 \pm 3\%$ in-river degradation.
Combined CSIA and mass balance analyses revealed a limited extent of in-river degradation, consistent with the short transit times from upstream sites to the main outlet, ranging from 2.4 hours to 8 days depending on discharge and location, with an average of $28 \pm 31$ hours (mean $\pm$ SD). These transit times are considerably shorter than the biodegradation half-life of *S*-metolachlor in the water column ($DegT_{50,oxic} = 29 \pm 8$ days) (Droz et al., 2021).





At the catchment scale, the extent of biodegradation estimated in October using CSIA ($98 \pm 20\%$, $\bar{x} \pm$ SD) aligned with the
overall biodegradation estimated by the mass balance ($99 \pm 5\%$, $\bar{x} \pm$ SD). Although CSIA-based estimates were subject to
considerable uncertainty due to analytical limitations, this was offset by the reduced sampling effort, smaller dataset
requirements, and fewer underlying assumptions compared to the mass balance approach.

In addition to estimating catchment-scale degradation, $\delta^{13}C$ measurements enabled to identify hotspots and hot moments of
pesticide degradation within the river network. Although limited to nine upstream-downstream $\delta^{13}C$ pairs from monthly grab
samples (Fig 3b, three in April and six in June), CSIA data revealed consistent degradation trends. In April, $\delta^{13}C$ values were
close to those of the applied commercial $S$-metolachlor (Fig. 3b; $\delta^{13}C_0 = -31.8 \pm 0.3‰$; Table S7 in the Supplement),
suggesting the export of non-degraded $S$-metolachlor into the river. By June, $\delta^{13}$C values indicated that approximately 60% of
the initial $S$-metolachlor mass had degraded, marking a degradation hot moment following its application. This rapid
degradation is consistent with the observed decline in $S$-metolachlor export, attributed to the reduced mass available for
transport from upstream sites (A1 and A2; high-frequency monitoring, Fig. 2d) to downstream (A3; Fig. 2g). $S$-metolachlor
transformation products further supported this degradation trend. Following spring application, a seasonal increase was
observed in the proportion of transformation products (sum of ESA, OXA, and NOA) relative to $MEL_{SM}$ (Eq. 1). Mean
transformation product proportions were $55.4 \pm 29.9\%$ (n = 26) in spring, $84.3 \pm 17.4\%$ (n = 16) in summer, $86.4 \pm 9.7\%$
(n = 22) in fall, and $95.6 \pm 2.9\%$ (n = 24) in winter (Section 2.8: Tab. S12 and Fig. S8 in the Supplement).

The limited number of upstream–downstream $\delta^{13}$C pairs did not reveal distinct degradation hotspots. This observation is
consistent with the 1:1 upstream–downstream concentration relationship (Fig. 3a), the uniform distribution of $S$-metolachlor
across corn and sugar beet fields (Fig. 1), and the relatively homogeneous lateral and hyporheic exchange along the river (see
Section 3.1).



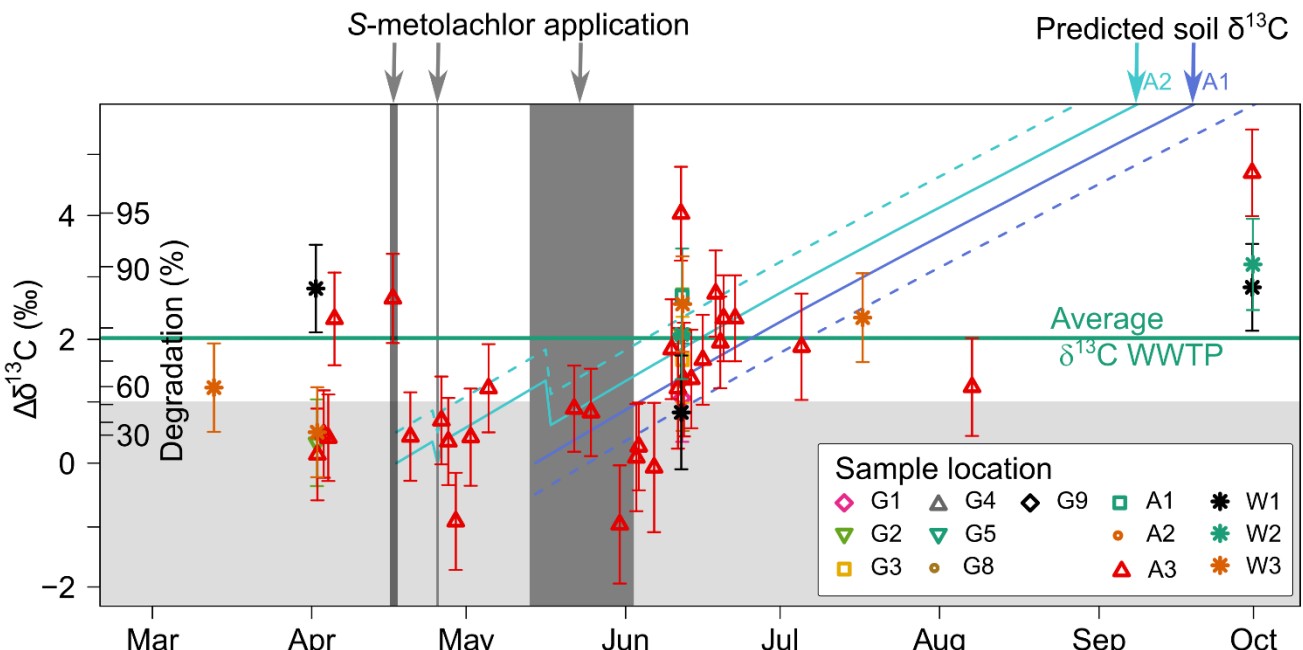

**Figure 4: Carbon stable isotope fractionation ($\Delta\delta^{13}C = \delta^{13}C(t) - \delta^{13}C_0$) of *S*-metolachlor in river (*n* = 61) water across the growing season compared to the predicted topsoil $\delta^{13}C$ values (Fig. S1 in the Supplement). The colored dashed lines indicate the uncertainty in the topsoil $\delta^{13}C$ values within ± 0.5‰. The shaded light grey area represents the minimal change in isotope signature ($\Delta\delta^{13}C_{min}$, calculated as per Eq. S1 in the Supplement) in water, beyond which significant isotope fractionation can be attributed to degradation, thereby representing the threshold for detecting biodegradation. The dark grey segment denotes the estimated date of *S*-metolachlor application ascertained from the farmer survey within sub-catchment G11. The green line represents the average $\delta^{13}C$ value from wastewater treatment plant data (*n* = 9). Error bars encompass the propagation of uncertainty associated with $\Delta\delta^{13}C$.**

The final application of CSIA, combined with a multi-scale sampling approach, focused on distinguishing the sources of *S*-metolachlor, specifically differentiating between WWTP and agricultural non-point pollution. From April to October, the isotope signature of *S*-metolachlor at the WWTP inlets remained consistent ($\delta^{13}C_{wwtp}$ = –29.76 ± 0.97‰; *n* = 9), showing an enrichment in $\delta^{13}C$ of 2‰ compared to the $\delta^{13}C$ value of the commercial formulation. This enrichment corresponds to approximately 80% *S*-metolachlor degradation (Fig. 4). Consequently, in rivers with substantial WWTP effluent contributions, an enriched $\delta^{13}C$ value of *S*-metolachlor early after the application may indicate a WWTP-derived source. Several mechanisms may explain the constant isotopic shift in *S*-metolachlor at WWTP inlets. These includes *S*-metolachlor degradation within the WWTPs (Gerecke et al., 2002) or the apportionment of *S*-metolachlor with an isotopic signature distinct from that of agricultural sources. However, further investigations are needed to precisely identify and attribute these sources, which would aid in developing effective mitigation measures.

Unlike the expected decline in $\delta^{13}C$ values from agricultural non-point sources following *S*-metolachlor application, the constant $\delta^{13}C$ values of WWTP-derived *S*-metolachlor can help differentiate these two sources. For instance, variations in





$\delta^{13}C$ exceeding 2‰ ($\Delta\delta^{13}C_{\text{riv}} = \delta^{13}C_{\text{downstream}} - \delta^{13}C_{\text{upstream}}$) in June for river reaches G6 → A3, G5 → A3 and G9 → A3 (Fig. 3b) cannot be solely be attributed to WWTP effluent. This indicates that *S*-metolachlor biodegradation likely occurred between downstream and upstream regions of the Souffel, or that the impact of a downstream point source emitting degraded *S*-metolachlor played a significant role.

However, the $\delta^{13}C$ values of point and non-point sources were very similar (Fig. 4), limiting the utility of CSIA for quantifying their individual contribution. Based on the mass balance, point sources associated with WWTP effluent contributed between 50% to over 80% of the *S*-metolachlor load at the catchment outlet throughout the season. This aligns with previous observations, such as those from the US, where *S*-metolachlor was detected in 74% of WWTPs and accounted for an estimated 47% of the total *S*-metolachlor mass in rivers (Sutton et al., 2019).

## 4 Conclusions and outlook

### 4.1 Wastewater treatment plant as a major river source of *S*-metolachlor

The integration of multi-scale sampling with a mass balance approach revealed an unexpected contribution of point sources, linked to WWTP effluent, to the overall *S*-metolachlor load at the catchment outlet. In response, collaboration between WWTP operators and agricultural advisors has initiated forensic investigations within the sewage network and awareness-raising efforts targeting farmers. These actions aim to improve understanding and mitigation of point-source contributions of *S*-metolachlor and other pesticides.

This study also underscores limitations of C-CSIA in distinguishing between point and non-point contributions to *S*-metolachlor loads. This limitation likely arises from the similarity of biodegradation processes occurring in both topsoil and in WWTPs, leading to similar degradation reactions and insufficient differences in $\delta^{13}C$ values of end members for isotopic mixing models to be effective. However, multi-element CSIA may potentially provide additional insights into differentiating *S*-metolachlor sources. For instance, sensitive analysis of isotopic fractionation of additional elements, in particular chlorine (Ponsin et al., 2019), may help if the cleavage mechanisms for these elements differ between WWTP degradation processes and those in topsoil or river environments. Currently, datasets characterising isotopic fractionation associated with the key pesticide degradation processes, such as biodegradation, photolysis, and hydrolysis, in WWTPs remain scarce, whereas they are better documented for topsoil and river environments (Drouin et al., 2021; Droz et al., 2021; Masbou et al., 2018). Further investigations into degradation pathways specific to WWTPs are essential to improve their capacity to mitigate micropollutants originating from urban metabolism. Such studies could enhance understanding of pesticide transformation in WWTPs and support the development of targeted measures to reduce pollutant loads in aquatic systems.



## 4.2 Potential of CSIA to evaluate *S*-metolachlor degradation in mid-scale catchment

This study demonstrates that C-CSIA at the catchment outlet effectively reflects the extent of *S*-metolachlor degradation in topsoil across scales ranging from a small sub-catchment (3.6 km²) to the entire catchment (115 km²). These findings are consistent with observations from small headwater catchments (e.g., 47 ha) (Alvarez-Zaldívar et al., 2018). A key advantage of CSIA is its independence from concentration data, mass balance calculations, and the need to identify transformation products, offering a significant advantage over other methods for estimating degradation extent. To expand CSIA applications to other compounds at the catchment scale, the stable isotope composition of active substances in pesticide formulations, such as $\delta^{13}C$, $\delta^{15}N$ or $\delta^{37}Cl$, should be characterised and compiled into accessible database similar as ISOTOPEST database (Masbou et al., 2024).

Reliable evaluation of degradation using C-CSIA alone requires a minimum isotope signature change ($\Delta\delta^{13}C_{min}$) of 2‰, which limits detection to biodegradation extents above 80% for *S*-metolachlor. However, recent advancement in chlorine isotope analysis, with its high isotopic fractionation for *S*-metolachlor ($\varepsilon_{Cl}$ = –9.70 ± 2.9‰) (Ponsin et al., 2019), may overcome these limitations, potentially enabling the sensitive detection of degradation extents lower than 50% in the future.

Another limitation of C-CSIA is the minimal carbon mass required for accurate GC-IRMS analysis. This constraint restricts its application to scenarios where sufficient residual pesticide remains to meet the 20-ng carbon requirement, typically corresponding to degradation extents below 95%. Despite this limitation, pre-concentration techniques, such as solid phase extraction (Gilevska et al., 2022), allowed reliable measurement of $\delta^{13}C$ values in river samples until October, corresponding to over 95% degradation.

Monthly grab sampling at the catchment outlet, combined with CSIA, proved to be an effective approach for monitoring ongoing degradation following pesticide application. This method reduces both monitoring and analytical costs compared to intensive, spatially and temporally distributed sampling campaigns, making it a practical option for agencies responsible for pesticide degradation monitoring. Nevertheless, high-frequency monitoring at the outlet remains essential for accurately quantifying pesticide export and evaluating reactive transport dynamics at the catchment scale.

A notable finding of this study is the limited contribution of the river to *S*-metolachlor degradation. This highlights the need for farmer advisors and water agencies to acknowledge the low natural attenuation capacity of rivers and prioritise measures to prevent pesticide entry into aquatic systems. For the Souffel river, a minimum in-river transit time of 17 days would be required to achieve significant degradation and associated $\delta^{13}C$ change. This limitation reduces the applicability of C-CSIA for assessing ongoing pesticide degradation in larger catchment (i.e., Strahler order >5) or systems with highly reactive river-sediment interfaces.

Altogether, this study shows that combining CSIA with multi-scale sampling enhances the ability to diagnose pesticide off-site transport and degradation, supporting the development of strategies to preserve and restore aquatic ecosystems. In particular, multi-element CSIA, including $\delta^{13}C$, $\delta^{15}N$ or $\delta^{37}Cl$ (Hofstetter et al., 2024; Höhener et al., 2022), may further help to evaluate the specific contribution of rivers to pesticide degradation and identify prevailing degradation pathways in

smaller Strahler order catchment. Multi-scale sampling across eleven river reaches provided valuable hydrological and chemical data, while high-frequency monitoring with nested catchments offered insights into the effects of losing and gaining streamflow conditions on *S*-metolachlor transport and reactivity. However, maintaining three stream gauge stations with associated sampling devices and weekly maintenance was proved challenging, therefore only the main outlet station was operational by the end of this study. Although *S*-metolachlor is no longer approved in the European Union due to its high leaching potential and risks to groundwater and human health (Commission Implementing Regulation (EU) 2023/1745), the framework developed in this study enables quantification of its transport and degradation as a legacy contaminant from topsoil to river networks in agricultural catchments. In summary, pesticide CSIA offers a promising approach for estimating degradation extents across entire catchments with minimal sampling and analytical effort. In the future, combining multi-scale monthly grab sampling with CSIA of both the parent compound and its associated transformation products could provide an effective approach for identifying degradation hotspots and hot moments within river networks, while simultaneously capturing associated hydrological and hydrochemical patterns.

## Data availability

The datasets for this study will be provided at Zenodo.

## Supplement

If the manuscript is accepted, the supplement related to this article will be available online at the HESS website.

## Author contributions

BD and GD performed conceptualisation, data curation, formal analysis, investigation and writing – original draft. JL performed formal analysis, investigation and writing – review & editing. BG performed conceptualisation, investigation and writing – review & editing. SP and GI performed conceptualisation, funding acquisition, project administration, supervision and writing – review & editing.

## Competing interests

The authors declare that they have no conflict of interest.



**Acknowledgements**

We gratefully acknowledge Margaret Johnson, Martine Trautmann, Colin Fourtet, Thierry Perrone, Eric Pernin, Jérémy Masbou, François Chabaux, Nolwenn Lesparre, Dimitri Rambourg, Agnès Herrmann, and Sylvain Benarioumlil (University of Strasbourg), as well as Laurent Mergnac (Syndicat des Eaux et de l'Assainissement Alsace-Moselle, SDEA), for their
contributions to both laboratory and field operations. We also extend our sincere thanks to Guillaume Ryckelynck (Région Grand Est), Blandine Fritsch (Chambre d'Agriculture Alsace), Marie Manceau and Rémy Gentner (Eurométropole Strasbourg), Quentin Morice (Direction Régionale de l'Environnement, de l'Aménagement et du Logement, DREAL), and the Mundolsheim city council for their essential support in coordinating field activities and providing spatial datasets.

**Financial support**

GD received financial support from the French Ministry for the Ecological Transition. BD was supported by a fellowship jointly funded by the Région Grand Est and the Rhine-Meuse. This research was funded by the Région Grand Est, the AERM (grant no. 170293), and the National School for Water and Environmental Engineering (École Nationale du Génie de l'Eau et de l'Environnement de Strasbourg, ENGEES).

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
