# Peer review of "How combining multi-scale monitoring and compound-specific isotope analysis helps to evaluate degradation of the herbicide S-metolachlor in agro-ecosystems?"

_EGUsphere, 2025_

## Referee Comment (RC1)

**Review of egusphere-2025-2309 –How combining multi-scale monitoring and compound-specific isotope analysis helps to evaluate degradation of the herbicide S-metolachlor in agro-ecosystems?**

The authors present an analysis of multi-scale monitoring in combination with CSIA to assess source apportionment and degradation of the herbicide S-metolachlor in a mesoscale agricultural catchment in France. They calculate that around 98% of S-metolachlor has been degraded over the 5-month growing season based on both their mass balance approach and CSIA. However, CSIA did not give a clear result regarding source apportionment between S-metholachlor from WWTPs and diffuse sources, respectively. The authors show that their CSIA methods can be a time and cost-efficient, yet reliable way to estimate pesticide degradation compared to a mass-balance approach requiring high-frequency measurements at different scales within a catchment.

**General and specific comments**

The study fits the scope of HESS. It makes a valuable contribution to the field of water quality and shows ways forward in catchment-scale monitoring. It is one of the first studies analysing the use of CSIA for assessing both sources and degradation of a pesticide. The authors present a substantial number of different methods and extensive monitoring data. Overall, the work is clear and described well in the paper. Thank you for the interesting read! See below for some specific comments on the text.

- The title: if you put it as a question, please add "does" or "can" for proper English grammar.
- Lines 76—77: These studies did not use isotope mixing models, at least if you think of the typical use of this term in literature. I'd prefer calling this isotope-fractionation reactive transport modelling, or reactive transport modelling including isotope fractionation or alike.
- Aim (i) of the study: "evaluate the potential of CSIA data collected along the river network as a proxy for evaluating upstream topsoil degradation of S-metolachlor". Only later in the text it becomes clear to me why you are not looking at CSIA in the topsoil directly, but rather use the in-stream isotope data as a proxy. The reader will only find out in the methods that the latter has not been the original aim, but is a result of the strong matrix effects in the analysis. I would leave this aim more open, as reading this immediately raised questions regarding why you would not go for the isotope data from the topsoil directly. So something along the lines of CSIA data collected at different locations to evaluate topsoil and river degradation of S-metolachlor.
- Figure 1: In this figure, A3 is not the outlet of the catchment. It is obvious that there will be little S-metolachlor added to the stream in the last bit, but I am wondering nonetheless why A3 does not coincide with the actual outlet of the catchment shown here.
- Line 141: could you mention to what extent these three transformation products cover all possible degradation pathways for S-metolachlor? This might be interesting in the comparison of mass balances via CSIA, transformation products, and Eq. 6.
- Lines 161-163: could you mention here what the range of timeframes covered by one composite sample is (roughly)? We can derive this from the data presented later, but it would be good to read about this here already.
- Equation 3: why do you use instantaneous and not average water discharges in period i, similar to what is done for the concentrations?

- Lines 286-287: "2019 for the same month was the five time drier and three times warmer,…". I do not understand – how did you determine this? What does three times warmer mean to you, for example?
- Lines 327-329: could you explain in more detail why the seasonal ESA to OXA ratio of 3.1"further supports the predominance of subsurface flow in the hydrological functioning of the Souffel catchment"? This is not clear to me.
- Lines 329-331: please correct this sentence.
- Figure 2:
  - Could you add the rainfall in panel (a) also on the right side?
  - What do the colours in d represent?
  - I think in general, the legend could be a bit more clear. It took me some time to understand what I am seeing here. Maybe better to have a small inset in each panel so we know right away what we are looking at?
- Lines 409-410: Could the authors briefly explain how they come to this finding? I assume it is because of the high export combined with low-intensity rainfall, but it would be good to explicitly mention this (briefly) here.
- Lines 443-444: This should be made more clear in the methods (Section 2.5.2). Otherwise it is not clear why d13C values need to be predicted for the topsoil, based on a model that quantifies biodegradation in the topsoil already.
- Lines 267-269: Why could be the main reason(s) that this is consistently lower than the 98% mass balance and CSIA-based estimates? Not all TPs accounted for? Or further breakdown (although they are thought to be persistent)? Not enough measurements?
- Figure 4: Why is there this difference in the predicted soil isotope values between A1 and A2? Metolachlor has not been applied in A1 before June? I am not sure whether this has been mentioned before.
- Lines 502-503: How are these numbers related to line 384 ("ranging from 0 to 100% of the observed mass load at the catchment outlet")?
- Lines 551-553: I am not sure I understand. Larger catchments with longer in-stream transit times or systems with highly reactive hyporheic zones would lead to more degradation and associated isotope fractionation. Would this not support the applicability of C-CSIA?

**Supplement**

- S1.5.2 Estimation of photodegradation in the Souffel river: I cannot follow the calculations easily, as some equations are not fully explained. What is I0? Is there a word missing in "Then, the depth of the photic zone in the Souffel River and the photic zone is defined as follows:"? What are absorbance and intensity used for? They do not reappear in (S4).
- S1.6 Variation of S-metolachlor concentrations in topsoil: Please clarify why which parameter is estimated. If I understand correctly, the S-metolachlor degradation is calculated to calculate isotope fractionation with (S9). That's why you get a straight line in Fig. S1. Why don't you use measured S-metolachlor concentrations in (S9)? Because of the limited number of samples?
- S1.9:
  - good to emphasize here (and also in the main text) that this is about in-stream transit times, not transit times through the subsurface
  - line 195: are you referring to Eq. (6)?

- Caption of Figure S3: "Colours get increasingly red later in the season,". This is not clear to me – the colour scale has been chosen this way.
- Table S12: Define that column TPS is indeed transformation products.

---

## Author Comment (AC1)

**Reviewer: Violaine Ponsin**

This study investigates the degradation of the widely applied herbicide S-metolachlor at the catchment scale through a nine-month field campaign. Two complementary approaches were employed: a conventional mass balance method based on extensive water and soil sampling and concentration measurements, and compound-specific isotope analysis (CSIA). Both methods gave consistent results, indicating that approximately 98% of the applied S-metolachlor is degraded during the growing season. The degradation occurs predominantly in topsoils, while in-stream degradation is limited, primarily due to short water residence times.

This manuscript provides a significant contribution to the field by showcasing the potential of CSIA in assessing pesticide dissipation under field conditions at the catchment scale, and is well presented. It clearly reflects an extensive effort and presents a large volume of data, although navigating the SI is not always straightforward.

The study is well aligned with the scope of HESS, but several comments must be addressed:

We sincerely thank you for your positive and constructive review. We greatly appreciate the time and effort you invested in carefully evaluating our manuscript, as well as your thoughtful identification of details, inaccuracies, and imprecisions that required clarification.

We have now addressed all your comments and suggestions line by line in the blue text below.

- 1) Isotope fractionation associated with biodegradation in topsoils had to be modeled due to matrix-related analytical issues (I assumed this refers to coelution or high background signal? It would be valuable to explicitly mention and maybe discuss these limitations, as they are relevant for the broader CSIA community).
  - It is unclear where the fractionation factor of −1.4 ‰ used in the model is coming from, as Droz et al. (2021) reported values of either −1.2 or −1.9 ‰. Clarification is needed.

The initial calculation used an isotopic enrichment factor of -1.2%, corresponding to oxic agricultural soil conditions in Droz et al 2021 (https://doi.org/10.1021/acs.est.0c06283). To account for the range of fractionation factors reported in the literature, including Torrento et al. (2021), as highlighted in the following comment, we have now incorporated these values into a revised model. The relevant references are now listed in the updated Table S7 (Supplement), and the associated uncertainty has been propagated throughout the model.

This revision is illustrated in the new Figure S1 and Figure 4, which present the median enrichment factor derived from the literature, along with its standard deviation ( $-1.84 \pm$

0.50‰ in Table S7). The changes have been implemented in both the *Materials and Methods* section (line 268-271) and the *Discussion* section (Fig4. line 500).

Additionally, Torrento et al. 2021 (https://doi.org/10.1021/acs.est.1c03981) reported a
fractionation factor for carbon of -2.4 % for S-metolachlor biodegradation in soils. It
would strengthen the analysis to include a sensitivity test or alternative model run using
this value for comparison.

As outlined in the previous point, the range of reported fractionation values is now explicitly incorporated into the model ( $-1.84 \pm 0.50\%$  in Table S7). This adjustment allows us to account for the sensitivity of the predicted  $\delta^{13}$ C values to this experimental parameter, particularly when comparing model outputs with observed  $\delta^{13}$ C values in the river. In response to this comment, we have now revised several sections of the manuscript, including:

L 268: "A median carbon isotopic enrichment factor for S-metolachlor ( $\varepsilon_{(bulk,C)} = -1.84 \pm 0.50\%$ ) was derived from laboratory-controlled biodegradation experiments conducted under a range of conditions reported in the literature (Table S7 in the Supplement). The contribution of photolysis to  $\varepsilon_{(bulk,C)}$  was considered negligible, based on findings by Van Breukelen (2007) (Sect. S1.8 in the Supplement)."

L 468-469: "the sensitivity of the CSIA method ( $\delta^{13}C = 1\%$ , <43% degradation)." It was <50% before considering the range of reported fractionation values.

L 476: "At the catchment scale, the extent of biodegradation estimated in October using CSIA (98  $\pm$  20%,  $\bar{x}$   $\pm$  SD) was consistent with the overall biodegradation estimated by the mass balance (99  $\pm$  5%,  $\bar{x}$   $\pm$  SD)." It was 98  $\pm$  2% before considering the range of reported fractionation values.

We have modified the caption (Lines 499 to 506): "Figure 4: Carbon stable isotope fractionation ( $\Delta\delta^{13}C = \delta^{13}C(t) - \delta^{13}C_0$ ) of S-metolachlor in river (n=61) water across the growing season compared to the predicted topsoil  $\delta^{13}C$  values. The colored dashed lines indicate the median uncertainty in the topsoil  $\delta^{13}C$  values within  $\pm$  0.5% (see Table S7 in the Supplement). The shaded light grey area represents the minimal change in isotope signature ( $\Delta\delta^{13}C_{min}$ , calculated as per Eq. S1 in the Supplement) in water, beyond which significant isotope fractionation can be attributed to degradation, thereby representing the threshold for detecting biodegradation. The dark grey segment denotes the estimated date of S-metolachlor application ascertained from the farmer survey within subcatchment G11. The green line represents the average  $\delta^{13}C$  value from wastewater treatment plant data (n=9). The black dash line represents the  $\Delta\delta^{13}C = 0\%$  value. Error bars encompass the propagation of uncertainty associated with  $\Delta\delta^{13}C$ ."

**And in the Supplement:**

**Lines 164 to 169:**

"Boundary values for biodegradation in soil reported in the literature and defined in this study range from  $\varepsilon_{biodeg,C}$  =  $-2.6 \pm 1.3\%$  to  $-1.2 \pm 0.4\%$  (Table S7; (Droz et al., 2021; Torrentó et al., 2021; Alvarez-Zaldívar et al., 2018; Meite, 2018). These values were used to determine the extent of degradation. The  $\delta^{13}C_0$  represents the isotopic signature of the commercial product (Table S8) (Alvarez-Zaldívar et al., 2018). The model enables the prediction of topsoil S-metolachlor concentrations and corresponding  $\delta^{13}C$  values at a higher temporal resolution than that achieved through monthly measurements.

Line 171: A new table S7: "

"Table S7: Range of isotopic enrichment factors for biodegradation ( $\varepsilon_{biodeg,C}$ ) of S-metolachlor reported in the literature for various soils and experimental conditions."

Line 176: A adapted figure S1

"Figure S1: Measured and predicted S-metolachlor topsoil concentrations at A1 and A2 (Fig. S2). Colored dashed lines represent the predicted uncertainty of the topsoil  $\delta^{13}C$  calculate using the uncertainty of the isotopic enrichment factors provided in Table S7."

2) Section SI 1.7 mentions that three piezometers were installed toward the end of the sampling campaign. Could the authors clarify the rationale behind this installation? What was the intended purpose, and how were the data used in the context of the study? Aside from a brief mention of groundwater electrical conductivity (P12L331), no groundwater data are presented or discussed.

The piezometer installed near the Avenheimerbach River was intended to clarify the influence of the shallow aquifer on river discharge and S-metolachlor loads. However, due to technical difficulties, installation was delayed until the end of the 2019 sampling campaign.

Given the limited value of the resulting dataset, this component of the sampling strategy has been removed from the revised manuscript (formerly lines 331–333). The corresponding Section S1.7 in the Supplementary Information has also been deleted.

**Specific comments**

P2L60-61: "tracking pesticide degradation under environmental conditions remains challenging due to limitations in current approaches." Approaches are described but their limitations are not.

We have now clarified the limitation of the current approaches as follows:

L 60-64: "Despite substantial efforts to study dissipation processes at the catchment scale, tracking pesticide degradation under environmental conditions remains highly challenging. Conventional approaches, based primarily on pesticide concentrations and the detection of transformation products (TPs), reflect both non-degradative dissipation and degradation across catchment compartments, but provide limited knowledge on the specific pathways and the extent of the degradation."

P3L65: "although its application has not previously been employed" please reformulate.

We have reformulated this sentence to place greater emphasis on the novelty of our study, as follows:

L.65 "In this context, compound-specific isotope analysis (CSIA) offers a promising tool for detecting and quantifying contaminant degradation in the environment (Elsner, 2010). However, its application to pesticide degradation in mid-scale agricultural catchments (from 50 to 500 km²) remains largely unexplored."

P3L69-70: enables and facilitates.

This has now been corrected.

P5L126: this approach was employed.

This has now been corrected.

P6L143: "water samples from eight monitoring sites". It is not clear to me whether these sampling points are those shown in Figure 1 (that shows nine sampling points), or different sampling points.

These eight sampling points were part of an independent dataset collected by the local water agency and are indicated in the figure S8 in the Supplement.

We have now clarified this aspect in the revised manuscript as follows, line L141: "An additional independent dataset from eight locations (Fig. S7 in the Supplement) provided by the Rhin-Meuse Water Agency was used to support evidence of S-metolachlor degradation."

P7L158: electrical conductivity.

The sentence has been corrected.

P11L286: please correct "for the same month was the five time drier" & P11L286-287: for every month, or just for some of them (in this case which ones)?

We agree that the original sentence lacked precision, as the comparison referred to the total discharge over the entire study period. Accordingly, L299-304 have been revised as follows:

"Compared to the previous twenty years (March to October 2000 to 2019; http://www.hydro.eaufrance.fr), the total discharge was five times lower with an areanormalized discharge at the outlet (A3) of  $0.828 \pm 0.986$  m3 day-1 ha-1 ( $\bar{x} \pm SD$ ; Fig. 2e). During the seven-month period from March to the end of September, 2019 was the fifth driest year in the past two decades, with total precipitation reaching 418  $\pm$  79 mm (2000–2019 average). In terms of temperature, it was also the third warmest year, with a mean of 16.3 °C, compared to the 2000–2019 average of 15.7  $\pm$  0.7 °C. These data are based on records from the Météo-France station in Entzheim, located approximately 10 km south of the catchment."

P11L298-299: Figure S6a doesn't really show that up to 100% of the flow comes from WWTP effluents during low-flow periods. Figure S6b does.

Thank you for this comment. Figure S5a (previously S6a) shows that in July and August, the combined effluents from the three WWTPs were approximately three times greater than the upstream river discharge. At the catchment outlet, river—groundwater interactions further reduce surface flow, allowing WWTP effluents to account for up to 100% of the observed discharge (Fig. S6). The influence of these effluents during low-flow periods was even more pronounced for S-metolachlor loads (Fig. S5b), due to the decline in diffuse upstream inputs approximately three months after the last herbicide applications, while point-source releases persist.

We have therefore revised the text to refer to Fig. S6 (previously S7) instead of Fig. S5a (previously S6a), in order to more accurately emphasize the role of WWTP effluents.

P12, L315-318: "During low-flow conditions, river—groundwater interactions further reduce surface water discharge, causing wastewater treatment plant (WWTP) effluents to contribute up to 100% of the total flow (Fig. S6 in the Supplement). This effect was even more pronounced for S-metolachlor loads (Fig. S5b in the Supplement), highlighting a sharp decline in diffuse upstream inputs roughly three months after the last herbicide applications, while point-source emissions from the WWTP continued."

P17L445-446: "However, due to a minimum carbon mass required for accurate GC-IRMS analysis ... only a subset of dissolved water samples (Fig. 4) was measurable." Does this introduce a bias in the reported isotope values, and is this a limitation of the CSIA approach in general? The lowest concentrations are often expected to exhibit the highest levels of degradation.

Out of context, the original sentence may be misleading. Here, we are referring to the instrumental detection limit for accurate measurements, specifically regarding the minimum amount that must be injected into the GC-IRMS system. This does not pertain to environmental concentrations.

We have therefore rephrased lines 465 to 467 as follows:

"However, accurate GC-IRMS analysis of S-metolachlor in the river water required a minimum amount of 20 ng of carbon per measurement. As a result, low-concentration samples lacking sufficient volume for preconcentration could not be analysed. Consequently, only a subset of in-stream samples (Fig. 4) met the criteria for reliable isotopic measurement."

With modification in the conclusion section 4.2 (lines 567 to 571):

"A key limitation of C-CSIA for S-metolachlor is the relatively high carbon mass required (i.e., 20 ng in this study) for accurate GC-IRMS measurement. This constraint may typically limit its applicability to scenarios where sufficient residual pesticide remains, generally corresponding to degradation extents below 95%. However, pre-concentration techniques such as solid-phase extraction (Gilevska et al., 2022) enabled reliable  $\delta^{13}$ C measurements in river samples through October, even when degradation exceeded 95%."

P18L454: The uncertainty associated with the extent of biodegradation estimated by CSIA is high compared to that obtained from the mass balance approach, and, according to the authors, this is due to analytical limitations. This point warrants further development.

We have now clarified the meaning of "analytical limitations" in L477 to L481 as follow: "The higher uncertainty associated with CSIA primarily stems from analytical challenges, such as detecting subtle isotope shifts near the instrumental detection limit, potential matrix interferences, and propagation of uncertainty inherent to Rayleigh-type modelling. Nevertheless, these limitations are partly counterbalanced by key advantages of CSIA, including lower sampling requirements, reduced data demands, and fewer assumptions relative to the mass balance approach."

Figure 4: it would be helpful to add the  $\Delta\delta 13C = 0$  % line.

We thank the reviewer for the suggestion to improve the figure's readability. In response, we have added a reference line at  $\Delta \delta^{13}C = 0\%$  to enhance interpretability.

P19L492: "the apportionment of S-metolachlor with an isotopic signature distinct from that of agricultural sources". This contradicts P15L390-395, which state that the most plausible

explanation for the occurrence of S-metolachlor in WWTPs is related to "releases during pesticide preparation ... or sprayer clean-out at farmyards".

We believe that the two sentences are not contradictory. In the first section (L390–395), we discussed the possible entry pathways of S-metolachlor into the WWTPs, identifying sprayer clean-out as the most plausible source. According to this hypothesis, the isotopic signature entering the WWTP should be close to that derived from the commercial products (–31.8‰) which are used (Table S8 in the Supplement). In the second sentence, we discussed the relatively stable isotopic values in the WWTP effluents (around –2‰ compared to initial signature). This stable isotopic signature in the WWTP effluents is likely associated with degradation processes occurring in the activated sludge of these WWTPs. Therefore, sprayer clean-out at farmyards, the most probable source of S-metolachlor entering the WWTPs, can be consistent with the relatively stable isotopic signature observed in the effluents.

We have rewritten Section L514-518, to clarify this point.

"A likely explanation is partial S-metolachlor degradation within WWTPs (Gerecke et al., 2002). Alternatively, some sources previously identified as plausible entry pathways, such as sprayer clean-out or runoff inputs, may introduce S-metolachlor with isotopic signatures slightly altered from those of the original commercial formulations. Disentangling these contributions will require further targeted investigation, which could support the development of more effective mitigation strategies"

P20L498-499: "This indicates that S-metolachlor biodegradation likely occurred between downstream and upstream regions of the Souffel". Again, this contradicts earlier discussions in the paper, which state that most of the degradation occurs in topsoils. Moreover, a substantial degree of degradation would be required to produce a measurable and significant shift in isotope values.

We believe that the two statements are not contradictory. In the first (P15, L390–395), we discuss potential entry pathways of S-metolachlor into the WWTPs, identifying sprayer clean-out as the most plausible source. Based on this hypothesis, the isotopic signature entering the WWTP should closely match that of the sprayer sample (-31.8%). The second paragraph does not contradict this interpretation but rather adds clarification by highlighting the comparatively limited degradation potential in the river relative to soil environments. To emphasise this point and strengthen the logical connection between the paragraphs, we have revised the second accordingly. two Consequently, lines 521 to 525 have been rewritten for clarity as follows: "The observed variations in  $\delta^{13}$ C between upstream and downstream sites suggest the influence of additional processes superimposed on the dominant topsoil degradation signal. These may include minor in-stream biodegradation, potentially occurring near the detection

limits of our CSIA method, or inputs from a downstream point source containing S-metolachlor that has already undergone partial degradation."

P20L519-520: "Currently, datasets characterising isotopic fractionation associated with the key pesticide degradation processes, such as biodegradation, photolysis, and hydrolysis, in WWTPs remain scarce". While this is generally true for many pesticides and micropollutants, it is less true for S-metolachlor, particularly concerning carbon isotopes. See for example Torrento et al., 2021 for hydrolysis and biodegradation (https://doi.org/10.1021/acs.est.1c03981) and Levesque-Vargas et al., 2025 for photodegradation (https://doi.org/10.1016/j.chemosphere.2024.144010). Although some processes are specific to WWTPs, others are not, and enrichment factors can probably still be applied beyond their original context.

This section now adopts a broader perspective on the application of Compound-Specific Isotope Analysis (CSIA) for tracking the degradation of organic pollutants. We have slightly reworded the text to reflect this more general approach. Accordingly, lines 544 to 550 have been revised as follows: "Comprehensive datasets on isotopic fractionation of herbicides linked to key degradation processes in the environment, such as biodegradation, photolysis, and hydrolysis, remain scarce. While fractionation values have been reported for some herbicides, including S-metolachlor and atrazine, their applicability beyond the original environmental conditions requires rigorous validation. Therefore, further research into herbicide degradation pathways specific to WWTPs is crucial for improving their effectiveness in mitigating micropollutants from urban sources. Such studies would advance our understanding of pesticide residues transformation within WWTPs and support the development of targeted strategies to reduce pollutant loads in aquatic systems."

P21L526-527 and L548: Water residence times in this catchment are very short. Would this statement remain valid in a watershed with ponds, where longer residence times are expected?

Our statement that "C-CSIA at the catchment outlet effectively reflects the extent of S-metolachlor degradation in topsoil across scales" remains valid, even for catchments with longer in-stream transit time and more advanced degradation. However, the sampling strategy, particularly the choice of water volumes, and the extraction and concentration steps using solid-phase extraction, should be optimised to enable the detection of more  $^{13}$ C-depleted  $\delta^{13}$ C values in water, which are indicative of lower residual S-metolachlor concentrations."

Accordingly, lines 578 to 583 have been revised as follows:

"In the Souffel River, a minimum in-stream transit time of approximately 17 days would be required to allow for significant degradation and a measurable  $\delta^{13}$ C shift. This constraint

illustrates why, in small agricultural catchments with short in-stream transit times and limited hyporheic reactivity, the use of C-CSIA to assess ongoing degradation is limited. In contrast, larger catchments (Strahler order >5) or systems with highly reactive riversediment interfaces typically exhibit longer in-stream transit times and stronger biogeochemical gradients, which can enhance degradation processes and increase the applicability of C-CSIA."

**Supporting information**

The table of contents should be more detailed to facilitate navigation.

The tables have now been more detailed to facilitate navigation.

The Y-axis unit should be revisited in Figure S1.

We have now revised the unit of the S-metolachlor concentration as µg per kg of soil.

Section S1.9: The text in this section suggests that fractionation factors for two different elements, C and N, were used in Equation S13 to derive a single effective fractionation factor (P10L191-192). This point requires clarification.

Thank you for this comment. The wording in Section S1.9 was indeed inaccurate and unintentionally implied that carbon and nitrogen fractionation factors were combined into a single effective factor in Equation S13. This was an oversight on our part. We have now corrected the section to clearly describe the distinct and intended application of the individual carbon and nitrogen fractionation factors.

We have removed the mention of nitrogen fractionation factors in the sentence (Lines 215-217):

"Contrasted isotopic enrichment factors as been observed for indirect photodegradation  $(\varepsilon_{photo,C}=0.0\pm0.0\%$  in Drouin et al. (2021) and  $-0.4\pm0.1\%$  in Levesque-Vargas et al. (2025)) and for biodegradation in soil ranging from  $\varepsilon_{biodeg,C}=-2.6\pm1.3\%$  to  $-1.2\pm0.4\%$  (Droz et al., 2021; Torrentó et al., 2021; Alvarez-Zaldívar et al., 2018; Meite, 2018)."

P15L281-282: Levesque-Vargas et al., 2025 reported isotopic enrichment factors for S-metolachlor photodegradation, although it was limited (-0.4  $\pm$  0.1‰ during indirect photodegradation)."

We thank the reviewer for highlighting the variability in isotopic fractionation values reported by different authors. We have evaluated both values in our calculation (see SI L216 to 225) and found that the range of isotopic enrichment factors reported in the literature for metolachlor photodegradation does not significantly affect the effective

isotopic fractionation  $(\varepsilon_{eff})$  when using the Van Breukelen method (https://doi.org/10.1021/es0628452) which account for the different kinetic and isotopic enrichment factor when multiple degradation pathway occur.

We have now clarified this issue in the main text L263

"The contribution of the photolysis on the  $\varepsilon_{bulk,C}$  has been estimated as negligible for S-metolachlor following Van Breukelen (2007) (Sect. S1.8 in the Supplement)."

as well as in the SI L219 to 225.

"As photodegradation and biodegradation co-occur in rivers, the Rayleigh Eq. (S9) was corrected according to Van Breukelen (2007) Eq. (S13):

$$\varepsilon_{eff,c} = \frac{k_{photo} \times \varepsilon_{photo,C} + k_{biodeg} \times \varepsilon_{biodeg,C}}{k_{photo} + k_{biodeg,C}} \approx \varepsilon_{biodeg,C}$$
(S13)

with  $\varepsilon_{eff}$  the effective isotopic enrichment factor to be re-injected in Eq. (6) to define the extent of in-stream degradation. However, the photodegradation term in Eq. S13 is negligible; consequently, in the present case study,  $\varepsilon_{eff,c}$  can be considered equivalent to  $\varepsilon_{biodeg,c}$ ."

Figure S6, X axis: it should be "sept" instead of "oct".

Thank you for identifying this; it has been corrected.

---

## Author Comment (AC2)

**Reviewer: Stefanie Lutz**

The authors present an analysis of multi-scale monitoring in combination with CSIA to assess source apportionment and degradation of the herbicide S-metolachlor in a mesoscale agricultural catchment in France. They calculate that around 98% of S-metolachlor has been degraded over the 5-month growing season based on both their mass balance approach and CSIA. However, CSIA did not give a clear result regarding source apportionment between S-metholachlor from WWTPs and diffuse sources, respectively. The authors show that their CSIA methods can be a time and cost-efficient, yet reliable way to estimate pesticide degradation compared to a mass-balance approach requiring high-frequency measurements at different scales within a catchment.

**General and specific comments**

The study fits the scope of HESS. It makes a valuable contribution to the field of water quality and shows ways forward in catchment-scale monitoring. It is one of the first studies analysing the use of CSIA for assessing both sources and degradation of a pesticide. The authors present a substantial number of different methods and extensive monitoring data. Overall, the work is clear and described well in the paper. Thank you for the interesting read! See below for some specific comments on the text.

• The title: if you put it as a question, please add "does" or "can" for proper English grammar.

Thank you for the contructive feedbacks and comments.

We have slightly modified the title as follow:

"How does integrating multi-scale monitoring and compound-specific isotope analysis improve the evaluation of S-metolachlor degradation in agro-ecosystems?"

• Lines 76—77: These studies did not use isotope mixing models, at least if you think of the typical use of this term in literature. I'd prefer calling this isotope-fractionation reactive transport modelling, or reactive transport modelling including isotope fractionation or alike.

We have revised lines 77 to 79 in accordance with the reviewer's suggestion and now use the term "isotope-fractionation reactive transport modelling".

"Moreover, isotope-fractionation reactive transport models based on CSIA data have been developed to support source apportionment at the hillslope scale (Lutz and Van Breukelen, 2014) and to predict pesticide biodegradation (Lutz et al., 2017)."

Aim (i) of the study: "evaluate the potential of CSIA data collected along the river network as a proxy for evaluating upstream topsoil degradation of S-metolachlor". Only later in the text it

becomes clear to me why you are not looking at CSIA in the topsoil directly, but rather use the instream isotope data as a proxy. The reader will only find out in the methods that the latter has not been the original aim, but is a result of the strong matrix effects in the analysis. I would leave this aim more open, as reading this immediately raised questions regarding why you would not go for the isotope data from the topsoil directly. So something along the lines of CSIA data collected at different locations to evaluate topsoil and river degradation of S-metolachlor.

To follow the suggestion, we have now rephrased the aim (L85-90):

"The study aimed to (i) evaluate the potential of CSIA data collected along the river network to evaluate topsoil and river degradation of S-metolachlor, (ii) quantify the river network contribution to overall degradation at the catchment scale, and (iii) differentiate between pesticide sources, including diffuse agricultural applications and point-source inputs from wastewater treatment plants (WWTP). To achieve these objectives, a multiscale sampling strategy was applied, integrating S-metolachlor mass balance, in-stream transit time analysis, and CSIA data from sources to the catchment outlet."

• Figure 1: In this figure, A3 is not the outlet of the catchment. It is obvious that there will be little S-metolachlor added to the stream in the last bit, but I am wondering nonetheless why A3 does not coincide with the actual outlet of the catchment shown here.

We apologise for the confusion. A3 represents the outlet of the study catchment. The original delineation shown in Figure 1 extended beyond the actual catchment boundary to include downstream land use for comparative purposes (see comparison below). To avoid further misunderstanding, we have revised Figure 1 by clipping the land use map to the actual catchment boundary.

• Line 141: could you mention to what extent these three transformation products cover all possible degradation pathways for *S*-metolachlor? This might be interesting in the comparison of mass balances via CSIA, transformation products, and Eq. 6.

Among the numerous transformation products (TPs) identified (typically over 30, as reported by Steele *et al.*, 2008, https://doi.org/10.2134/jeq2007.0166), metolachlor ESA, OXA, and NOA are consistently described as the most prevalent in environmental waters, both in terms of frequency of detection and concentration (https://doi.org/10.1016/j.scitotenv.2022.156696,

https://doi.org/10.1021/acs.est.1c00466 and https://doi.org/10.1007/s11356-025-35979-3) and they have legal thresholds across for groundwater (European Commission , Off. J. Eur. Union, L 155, 127-175, 2011). These three TPs of metolachlor are those regulated in Europe (EU Regulation No. 546/2011).

We incorporated an independent dataset provided by the Rhin-Meuse Water Agency as supplementary evidence of ongoing degradation processes over the same period and in locations proximal to our sampling sites. However, the monthly grab-sampling protocol employed by the agency precluded accurate integration of TP concentrations into our mass-balance calculations. This limitation has been clarified in the revised manuscript (L145–L148):

"These three TPs do not encompass all possible degradation pathways of S-metolachlor; instead, they represent a subset of particular concern for groundwater used as a drinking resource. This concern arises from their high mobility, persistence, and are frequent detection across Europe (Baran et al., 2022; Menger et al., 2021; Pasquini et al., 2025). Moreover, each is subject to regulatory thresholds for groundwater quality (European Commission, 2011)."

And lines 155-157: "However, the monthly grab sampling protocol employed by the Water Agency, conducted without simultaneous discharge measurements, precluded accurate integration of TP concentrations into the mass balance calculations."

• Lines 161-163: could you mention here what the range of timeframes covered by one composite sample is (roughly)? We can derive this from the data presented later, but it would be good to read about this here already.

Water samples, associated to flow proportional sampling were collected weekly and combined into composite samples. These composite samples represented water collected over a period ranging from 4.9 to 21 hours (22.8  $\pm$  0.8 hours,  $\bar{x} \pm$  SD), determined according to hydrograph characteristics that distinguished between baseflow and high-flow conditions.

We have now incorporated this information in lines 167–170.: "Water samples (up to twelve 330 mL glass bottles) were collected weekly and merged into composite samples, represented water collected over a 4.9 to 21-hour period (22.8  $\pm$  0.8 hours,  $\bar{x}$   $\pm$  SD). The sampling window was determined based on hydrograph characteristics, allowing separation of baseflow and high-flow conditions."

• Equation 3: why do you use instantaneous and not average water discharges in period i, similar to what is done for the concentrations?

We have clarified this in Line 232 to L236: "S-metolachlor concentrations were obtained from composite water samples, representing period-averaged values. In contrast, discharge data were available at high temporal resolution and used in their instantaneous form. By pairing these representative concentrations with continuous flow measurements, we accounted for temporal variability in hydrological conditions, enabling a more accurate estimate of total pesticide mass export over the monitoring period, rather than isolated load snapshots."

• Lines 286-287: "2019 for the same month was the five time drier and three times warmer,...". I do not understand – how did you determine this? What does three times warmer mean to you, for example?

The sentence has been corrected as follows (line 301 to 304): "During the seven-month period from March to the end of September, 2019 was the fifth driest year in the past two decades, with total precipitation reaching 418  $\pm$  79 mm (2000–2019 average). In terms of temperature, it was also the third warmest year, with a mean of 16.3 °C, compared to the 2000–2019 average of 15.7  $\pm$  0.7 °C. These data are based on records from the Météo-France station in Entzheim, located approximately 10 km south of the catchment."

Lines 327-329: could you explain in more detail why the seasonal ESA to OXA ratio of 3.1 further supports the predominance of subsurface flow in the hydrological functioning of the Souffel catchment"? This is not clear to me.

Rose et al. (2018, https://doi.org/10.1016/j.scitotenv.2017.08.154) identified distinct patterns of ESA/OXA ratios according to the dominant hydrological pathway, i.e. runoff- or subsurface flow-dominated discharge, across a set of catchments. With an ESA/OXA ratio of 3.1, the Souffel catchment is classified, for the seven-month period in 2019, as subsurface flow-controlled, in accordance with the typology proposed by Rose et al. (see "Fig. 3. Mean percent MET, MESA and MOXA in water samples from the environmental compartments in the seven study areas. The shaded areas represent the estimated expected ratio of metolachlor to degrade based on an environmental compartment's flow path" in Rose et al., 2018). We have now clarified the link with Rose et al. conclusion, lines 345-348:

"The seasonal ESA/OXA ratio of 3.1, derived from mean concentrations of  $56.8 \pm 12.3\%$  for ESA and  $16.7 \pm 4.4\%$  for OXA across the eight Water Agency sites (n = 88; Table S13 in the Supplement), aligns with the distinct ratio patterns reported by Rose et al. (2018), which are indicative of dominant hydrological pathways. Specifically, this ratio supports the

predominance of subsurface flow in the hydrological functioning of the Souffel catchment."

• Lines 329-331: please correct this sentence.

We have re-written the sentence as in L349-350: "During dry periods, higher electrical conductivity was observed at site A2 ( $\sigma$  = 1.19 ± 0.25 mS cm-1) compared with site A1 ( $\sigma$  = 0.58 ± 0.31 mS cm-1;  $\bar{x}$  ± SD), suggesting mixing between groundwater and surface water."

Figure 2: Could you add the rainfall in panel (a) also on the right side?

What do the colours in d represent?

I think in general, the legend could be a bit more clear. It took me some time to understand what I am seeing here. Maybe better to have a small inset in each panel so we know right away what we are looking at?

We thank the reviewer for the constructive suggestions regarding Figure 2. In response, we have revised the figure so that each panel now includes its own inset legend. We believe this substantially improves clarity compared to the previous single shared legend, allowing readers to more readily interpret the data.

Regarding the suggestion to add rainfall data to the right side of panel (a), we carefully considered this option. While we acknowledge the potential benefit of positioning rainfall directly above the discharge panel, duplicating the rainfall plot on both sides could create confusion, potentially leading readers to infer differences or additional information between the panels.

To avoid this ambiguity, we have opted to retain a single rainfall panel. We note, however, that major rainfall events are already indicated by dashed lines superimposed on the discharge curve, providing a concise and integrated visual reference without redundancy.

Lines 409-410: Could the authors briefly explain how they come to this finding? I assume it is because of the high export combined with low-intensity rainfall, but it would be good to explicitly mention this (briefly) here.

As Events 1 to 7 occurred shortly after S-metolachlor applications on sugar beet and maize plots, and a substantial residual mass was predicted to remain following these events (see Fig. S1 in the Supplement), we can reasonably conclude that the initial phase of S-metolachlor load dynamics was transport-limited. In contrast, the later phase, observed in July and August, was mass-limited, consistent with significant in situ biodegradation in the topsoil. A comparable shift from transport-limited to mass-limited behaviour over the

course of a growing season has been reported for various pesticides in a vineyard catchment (Imfeld et al., 2020).

In the revised draft, we provide a more detailed explanation of this process in Lines 425 to 431 as follows: "Given the high and spatially uniform S-metolachlor load across the catchment (Fig. 1; Table S1 in the Supplement) and the timing of herbicide application, the mass reservoir during events 1 to 7 (up to 15 June) can be considered effectively semi-infinite relative to runoff volumes. This implies that once hydrological connectivity was established, S-metolachlor was readily mobilised (Stieglitz et al., 2003). Consequently, pesticide export was transport-limited, accounting for 91% of the total seasonal load. As the season progressed and S-metolachlor stores in the catchment were depleted, source dynamics shifted to a mass-limited regime, consistent with previous observations (Peter et al., 2020; Fairbairn et al., 2016)."

Lines 443-444: This should be made more clear in the methods (Section 2.5.2). Otherwise it is not clear why d13C values need to be predicted for the topsoil, based on a model that quantifies biodegradation in the topsoil already.

This section has now been rewritten (Lines 274 to 279) to address a similar comment from the Reviewer #1. The revised section now reads as follow:

"Therefore, degradation dynamics and extent were inferred from monthly S-metolachlor concentration data (Section 2.3.2) using a degradation model that relates dissipation, degradation, and stable isotope fractionation. This model, previously validated in a nearby agricultural headwater catchment with similar soils and farming practices (Sect. S1.6 in the Supplement; Payraudeau et al., 2025), was applied to independently simulate topsoil S-metolachlor concentrations and corresponding  $\delta^{13}C$  values locations A1 and A2 with higher temporal resolution. Modelled S-metolachlor concentrations were then validated against observed topsoil measurement (Fig. S1 in the Supplement)."

Lines 267-269: Why could be the main reason(s) that this is consistently lower than the 98% mass balance and CSIA-based estimates? Not all TPs accounted for? Or further breakdown (although they are thought to be persistent)? Not enough measurements?

We assume that the targeted lines are Lines 467 to 469. "Following spring application, a seasonal increase was observed in the proportion of transformation products (sum of ESA, OXA, and NOA) relative to  $MEL_{SM}$  (Eq. 1). Mean transformation product proportions were  $55.4 \pm 29.9\%$  (n = 26) in spring,  $84.3 \pm 17.4\%$  (n = 16) in summer,  $86.4 \pm 9.7\%$  (n = 22) in fall, and  $95.6 \pm 2.9\%$  (n = 24) in winter (Section 2.8: Tab. S12 and Fig. S8 in the Supplement)."

Considering the upper boundaries of the contribution of three main transformation products, i.e. ESA, OXA and NOA on MELSM, with 86+9.7% in fall and 95.6 + 2.9% in winter, are not so far to the estimated contribution of degradation close to 98% of the applied S-metolachlor. However, the monthly grab sampling protocol employed by the Water Agency, without corresponding discharge measurements, prevented the accurate integration of TP concentrations into our mass balance calculations.

We have clarified this limitation in relation to the mass-balance analysis in Lines 155–157. "However, the monthly grab sampling protocol employed by the Water Agency, conducted without simultaneous discharge measurements, precluded accurate integration of TPs concentrations into the mass balance calculations."

Figure 4: Why is there this difference in the predicted soil isotope values between A1 and A2? Metolachlor has not been applied in A1 before June? I am not sure whether this has been mentioned before.

The S-metolachlor isotopic signature in topsoil was predicted using the modelling approach developed by Payraudeau et al. 2025 (https://doi.org/10.5194/hess-29-4179-2025). Degradation and the associated isotopic signature differ between A1 and A2, reflecting the reported applications and the hydro-climatic dynamics of the topsoil, which explains the differences between the predicted signatures in A1 and A2 topsoil. In the new figure 4, we have averaged the predicted soil isotope values from these two soils and integrated the range of enrichment factor derived from the literature, considering the associated mean and standard deviation ( $-1.84 \pm 0.50\%$  in Table S7).

Accordingly, we have modified the caption (Lines 498 to 505: "Figure 4: Carbon stable isotope fractionation ( $\Delta\delta^{13}C = \delta^{13}C(t) - \delta^{13}C_0$ ) of S-metolachlor in river (n= 61) water across the growing season compared to the predicted topsoil  $\delta^{13}C$  values. The colored dashed lines indicate the median uncertainty in the topsoil  $\delta^{13}C$  values within  $\pm 0.5\%$  (see Table S7 in the Supplement). The shaded light grey area represents the minimal change in isotope signature ( $\Delta\delta^{13}C_{min}$ , calculated as per Eq. S1 in the Supplement) in water, beyond which significant isotope fractionation can be attributed to degradation, thereby representing the threshold for detecting biodegradation. The dark grey segment denotes the estimated date of S-metolachlor application ascertained from the farmer survey within sub-catchment G11. The green line represents the average  $\delta^{13}C$  value from wastewater treatment plant data (n= 9). The black dash line represents the  $\Delta\delta^{13}C = 0\%$  value. Error bars encompass the propagation of uncertainty associated with  $\Delta\delta^{13}C$ ."

Lines 502-503: How are these numbers related to line 384 ("ranging from 0 to 100% of the observed mass load at the catchment outlet")?

We thank the reviewer for this detailed comment and for highlighting the lack of clarity in our statement. The two lines are related: Line 399 specifically refers to the monthly discrete observations at particular time points, whereas Lines 527–528 present the estimates based on mass balance calculations, integrating all sampling days over the season.

In Line 399, we have modified the sentence to specify that it represents the discrete daily load observed on a monthly basis.

"The proportion of WWTP derived S-metolachlor ( $x_{wwtp}$ ) varied throughout the season (Fig. S6b in the Supplement), ranging from 0 to 100% of discrete daily load observed on a monthly basis at the catchment outlet, with an average contribution of 53% from March to June."

Lines 527 to 528 have been rewritten for greater clarity as follows: "Mass balance calculations integrating all sampling days over the season indicate that WWTP effluent contributed between 50% and over 80% of the S-metolachlor load at the catchment outlet."

Lines 551-553: I am not sure I understand. Larger catchments with longer in-stream transit times or systems with highly reactive hyporheic zones would lead to more degradation and associated isotope fractionation. Would this not support the applicability of C-CSIA?

We have clarified in lines 577 to 582 that our catchment represents a specific case with limited suitability for applying CSIA to isolate the role of in-stream degradation. We acknowledge, however, that under different hydrological or biogeochemical conditions such as in larger catchments with longer in-stream transit times—C-CSIA can be highly effective. Lines have 577 to 582 been revised as follows: "In the Souffel River, a minimum in-stream transit time of approximately 17 days would be required to allow for significant degradation and a measurable  $\delta^{13}C$  shift. This constraint illustrates why, in small agricultural catchments with short in-stream transit times and limited hyporheic reactivity, the use of C-CSIA to assess ongoing degradation is limited. In contrast, larger catchments (Strahler order >5) or systems with highly reactive riversediment interfaces typically exhibit longer in-stream transit times and stronger biogeochemical gradients, which can enhance degradation processes and increase the applicability of C-CSIA."

**Supplement**

• S1.5.2 Estimation of photodegradation in the Souffel river: I cannot follow the calculations easily, as some equations are not fully explained. What is IO? Is there a word missing in "Then, the depth of the photic zone in the Souffel River and the photic zone is defined as follows:"? What are absorbance and intensity used for? They do not reappear in (S4).

We apologise for the lack of clarity in this section. We have carefully reorganized the section and rephrased passages whenever necessary. We believe that Section S1.5.2 in the supplementary Information is now clearer and reads more smoothly.

• S1.6 Variation of S-metolachlor concentrations in topsoil: Please clarify why which parameter is estimated. If I understand correctly, the S-metolachlor degradation is calculated to calculate isotope fractionation with (S9). That's why you get a straight line in Fig. S1. Why don't you use measured S-metolachlor concentrations in (S9)? Because of the limited number of samples?

Topsoil samples (0 to 10 cm) were collected monthly from sugar beet and corn fields at two locations within the surveyed sub-catchment (within a 100 m radius of sites A1 and A2, Fig. 1) to quantify S-metolachlor dissipation over the season. However, isotope fractionation associated with biodegradation in topsoils had to be modeled due to matrix-related analytical issue, using the modelling approach developed by Payraudeau et al. 2025 (https://doi.org/10.5194/hess-29-4179-2025), reflecting the reported applications and the hydro-climatic dynamics of the topsoil. This model, validated in a nearby agricultural headwater catchment with comparable soils and farming practices (Sect. S1.6 in the Supplement; Payraudeau et al., 2025), was then applied to predict topsoil S-metolachlor concentrations and the corresponding  $\delta^{13}$ C values at daily temporal resolution for soil locations A1 and A2 independently (Fig. 1). The predicted topsoil S-metolachlor concentrations were validated against the observed concentrations (Fig. S1 in the Supplement).

We have clarified this topsoil modelling step in the revised manuscript (Lines 274 to 279):

"Therefore, degradation dynamics and extent were inferred from monthly S-metolachlor concentration data (Section 2.3.2) using a degradation model that relates dissipation, degradation, and stable isotope fractionation. This model, previously validated in a nearby agricultural headwater catchment with similar soils and farming practices (Sect. S1.6 in the Supplement; Payraudeau et al., 2025), was applied to independently simulate topsoil S-metolachlor concentrations and corresponding  $\delta^{13}C$  values locations A1 and A2 with higher temporal resolution. Modelled S-metolachlor concentrations were then validated against observed topsoil measurement (Fig. S1 in the Supplement)."

S1.9: good to emphasize here (and also in the main text) that this is about in-stream transit times, not transit times through the subsurface

We have replaced *transit time* with *in-stream transit time* throughout Section S1.9 and the main text.

• Caption of Figure S3: "Colours get increasingly red later in the season,". This is not clear to me –the colour scale has been chosen this way.

The colour scale corresponds to the date and follows the legend presented in the figure. The colour scheme was selected in accordance with the journal's guidelines and to ensure accessibility for readers with colour vision deficiencies. We carefully evaluated the chosen colours using the Coblis – Colour Blindness Simulator and revised the colour scheme accordingly.

• Table S12: Define that column TPS is indeed transformation products.

TPs and other acronyms have now been clearly defined in the legend of the table S12.